# A New Natural Defense Against Airborne Pathogens

Covid-19; Aerosols; Exhaled Particles; Hygiene; Nasal Delivery; Saline

**Author for correspondence:**
David Edwards,
E-mail: Dedwards@seas.harvard.edu

David Edwards[1,2] (iD), Anthony Hickey[3,4], Richard Batycky[5], Lester Griel[6], Michael Lipp[5], Wes Dehaan[7], Robert Clarke[8], David Hava[8], Jason Perry[8], Brendan Laurenzi[8], Aidan K. Curran[8], Brandon J. Beddingfield[9], Chad J. Roy[9], Tom Devlin[2] and Robert Langer[10]

[1]Harvard John A. Paulson School of Engineering & Applied Sciences, Harvard University, Cambridge MA, USA; [2]Sensory Cloud Inc., Boston, MA, USA; [3]RTI International, Research Triangle Park, NC, USA; [4]Eshelman School of Pharmacy, University of North Carolina at Chapel Hill, Chapel Hill, NC, USA; [5]Nocion Therapeutics, Waltham, MA, USA; [6]Department of Veterinary and Biomedical Sciences, Penn State University, State College, PA, USA; [7]Selecta Biosciences, Watertown, MA, USA; [8]Pulmatrix Inc., Lexington, MA, USA; [9]Department of Microbiology and Immunology, Tulane School of Medicine, New Orleans, LA, USA and [10]Department of Chemical Engineering, Massachusetts Institute of Technology, Cambridge, MA, USA

## Abstract

We propose the nasal administration of calcium-enriched physiological salts as a new hygienic intervention with possible therapeutic application as a response to the rapid and tenacious spread of COVID-19. We test the effectiveness of these salts against viral and bacterial pathogens in animals and humans. We find that aerosol administration of these salts to the airways diminishes the exhalation of the small particles that face masks fail to filter and, in the case of an influenza swine model, completely block airborne transmission of disease. In a study of 10 human volunteers (5 less than 65 years and 5 older than 65 years), we show that delivery of a nasal saline comprising calcium and sodium salts quickly (within 15 min) and durably (up to at least 6 h) diminishes exhaled particles from the human airways. Being predominantly smaller than 1 μm, these particles are below the size effectively filtered by conventional masks. The suppression of exhaled droplets by the nasal delivery of calcium-rich saline with aerosol droplet size of around 10 μm suggests the upper airways as a primary source of bioaerosol generation. The suppression effect is especially pronounced (99%) among those who exhale large numbers of particles. In our study, we found this high-particle exhalation group to correlate with advanced age. We argue for a new hygienic practice of nasal cleansing by a calcium-rich saline aerosol, to complement the washing of hands with ordinary soap, use of a face mask, and social distancing.

## Introduction

Airborne transmission of infectious disease by the very small droplets we emit from our airways on natural breathing, and that accumulate in poorly ventilated indoor environments, has been observed for a range of respiratory diseases, including tuberculosis, measles, chicken pox, influenza and severe acute respiratory syndrome (SARS) (Li *et al.*, 2007). Recent observations (Zhang *et al.*, 2020) place the novel coronavirus SARS-CoV-2—carried by the small airborne droplets exhaled by COVID-19 infected individuals (Liu *et al.*, 2020) and reported stable in aerosol form (Van Doremalen *et al.*, 2020) for beyond 3 h—in the family of airborne transmitted infectious diseases as well. The assessment of SARS-CoV-2 as an airborne pathogen clarifies the nature of the fight against the COVID-19 pandemic (Fauci *et al.*, 2020).

While major international scientific efforts advance toward the development of drugs and vaccines in response to the COVID-19 pandemic, less attention has been given to new ways to prophylactically combat airborne viral and bacterial threats. Social distancing, washing of hands, and wearing of face masks, while each effective and necessary, do not prevent the transmission of pathogens that travel through the air within droplets smaller than 1 μm in diameter. Such small particles are not only too small to be filtered effectively by conventional masks (Bunyan *et al.*, 2013; Leung *et al.*, 2020; Zhang *et al.*, 2020), but also too small to settle by gravity within the 2-m threshold of social distancing.

Submicron droplets happen to be the majority of particles we emit from our mouths and noses when we naturally breathe. They emerge from our respiratory systems either by the necking of airway lining fluid (ALF) that occurs with the expansion and contraction of the lungs (Scheuch, 2020), or by the rapid movement of air through upper airways, as occurs during natural breathing, coughing, sneezing, and speaking (Watanabe *et al.*, 2007). Whether high or low in the airways, shedding of small droplets from infected lungs can carry viral and bacterial

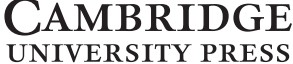

pathogens into the environment, promoting disease spread (Edwards *et al.*, 2004; Leung *et al.*, 2020). When this shedding occurs in the upper airways, it can promote movement of pathogen deeper into the lungs, and self-infection.

The delivery of saline to the respiratory system has been observed to diminish the exhalation of these very small particles (Edwards *et al.*, 2004). This diminution is due to electrostatic interactions between salt cations and mucin and mucin-like proteins on the surface of ALF. These increase the viscoelasticity of the surface of ALFs (Watanabe *et al.*, 2007), reducing the formation of airborne droplets (Watanabe *et al.*, 2007). Calcium chloride ($CaCl_2$), far more than sodium chloride (NaCl), has been found to increase the surface elasticity of ALF, potentially promoting even more substantial suppression of airborne particle generation in the airways, while enhanced surface plasticization also further resists pathogen penetration through the mucus layer, strengthening its biophysical barrier to infection (Watanabe *et al.*, 2007).

Calcium and sodium salts appear to have antimicrobial properties as well. High concentrations of extracellular calcium (Krisanaprakornkit *et al.*, 2003), as can be achieved by the aerosol delivery of calcium salts, promote the secretion of β-defensin 2 from nasal epithelial cells (Alp *et al.*, 2005). Human β-defensin 2 is an endogenous antimicrobial peptide that conjugates with receptor-binding domains of many viruses, including coronaviruses, to promote expression of antiviral and immune-inducing molecules as well as chemokine recruiters of leukocytes (Kim *et al.*, 2018). Human β-defensin 2 has been shown to be effective as an antiviral adjuvant due to its binding to the spike protein of Middle East respiratory syndrome coronavirus (MERS-CoV) (Zhao *et al.*, 2016), and mouse beta defensin 4 derived peptide has shown activity against SARS-CoV-1 (Kim *et al.*, 2018).

Chloride salts have been shown to diminish viral replication as far back as the 1960s (Speir, 1961). Chloride ions promote antiviral activity by the induction within cells of hypochlorous acid, the active constituent of bleach (Ramalingam *et al.*, 2018). Chloride salts induce innate immune response of epithelial cells in the presence of NaCl (Ramalingam *et al.*, 2018). Sodium hypochlorite, the sodium salt of hypochlorous acid, has particularly demonstrated effectiveness as a disinfectant against coronavirus (Dellanno *et al.*, 2009; Geller *et al.*, 2012). High concentrations of chloride, delivered via hypertonic saline to nasal epithelial tissues, have been found to diminish viral infections associated with the common cold (Adam *et al.*, 1998; Šlapak *et al.*, 2008; Ramalingam *et al.*, 2019). In an open-labeled randomized controlled human study of 68 subjects with common cold infections including rhinoviruses and coronaviruses, as well as enterovirus and influenza A virus, nasal delivery of 2–3% hypertonic saline 2–8 times a day (median thrice-a-day) significantly lowered duration of illness, as well as use of over-the-counter medications, household transmissions, and viral shedding (Ramalingam *et al.*, 2019). Nasal administration of 3.5% hypertonic seawater has similarly shown indications of efficacy against common cold symptoms in other human clinical trials (Adam *et al.*, 1998; Šlapak *et al.*, 2008).

To address the need for a broad prophylactic and anti-contagion defense against respiratory viral and bacterial infections, we designed salt compositions for hygienic and therapeutic applications by incorporating three ions that are abundant in human tissues: calcium, sodium, and chloride. We hypothesized that an aerosol combining calcium and sodium salts would improve the barrier function of the mucus lining to protect against infection and promote natural innate immune disinfectant properties suited to a range of bacterial and viral infections, while diminishing bioaerosol formation in the lungs

and nasal passages. Given the hygienic practice of nasal salines, we decided to evaluate the nasal delivery of our physiological saline with a specially designed nasal mist, as a practical, efficacious, and safe personal hygiene intervention, complementary to masks. We assumed that this might prove of particular utility to the immediate fight against the current COVID-19 pandemic.

## Results

### Compositions and cell models: influenza, rhinovirus, and pneumonia

We sought to establish optimal compositions and mechanism of action for physiological salts suited to antimicrobial activity against viral and bacterial pathogens with in vitro systems. Formulations were prepared with compositions across a range of $CaCl_2$ and NaCl ratios.

These formulations were aerosolized by nebulizer and delivered onto air–liquid interface (ALI) cultures formed by Calu3 cells (a human cultured epithelial cell line) prior to infection with influenza A/WSN/33/1. To assess antimicrobial transport properties, other experiments were performed by nebulizing the formulations onto ALF mimetic systems comprised of 4% sodium alginate using viral (influenza A/WSN/33/1 or rhinovirus [Rv16]) and a range of Gram-positive and Gram-negative bacterial (*Streptococcus pneumonia* [Sterotype 4; TIGR4], *Klebsiella pneumonia* [ATCC 43816], *Pseudomonas aeruginosa* [PAO1], *Staphylococcus aureus* [ATCC 25923], and non-typeable *Haemophilus influenza* [14P14H]) pathogens. In order to identify compositions of optimal efficacy and clarify mechanism of action, we evaluated the ability of these formulations to lower viral titers collected (in the case of the Calu3 ALI model) in an apical wash 24 h after administration and (in the case of the ALF mimetic model) to lower rate of progression across ALF.

We found that exposing the Calu3 cells via nebulization to solutions of 1.29% (0.115 M) $CaCl_2$ or 12.9% (1.15 M) $CaCl_2$ in saline (0.9% NaCl) reduced viral titers in a $CaCl_2$ dependent manner (Fig. 1*a*). This study was repeated with magnesium chloride ($MgCl_2$) to determine whether increased tonicity or specific ionic composition determined efficacy; the $MgCl_2$ was dissolved in saline at identical molar concentrations as the $CaCl_2$. As shown in Fig. 1*b*, viral titers did not significantly change with $MgCl_2$ exposure, suggesting calcium, more than chloride or sodium, was most responsible for the reduction in viral titer. We further examined relative contributions across a range of tonicities. The results, shown in Fig. 1*c*, show that ion type and not tonicity itself is critical to effectiveness, and that calcium, with a strong synergistic effect of NaCl, was the primary active ingredient. $CaCl_2$ alone was also less effective, suggesting perhaps a secondary chloride role, with chloride concentration titrated independently of calcium by sodium.

We sought to determine optimal ratios of $CaCl_2$ and NaCl based on antiviral efficacy through a series of experiments with the Calu3 ALF influenza model. This led to the identification of three optimal compositional zones as shown in Fig. 1*d*. We chose in our subsequent studies to explore three unique (fast emergency nasal defense, FEND) composition ranges of $CaCl_2$ and NaCl, all with hypertonicity ranges well tolerated in the pulmonary treatment of cystic fibrosis. The constituents ($CaCl_2$ and NaCl) and tonicity ranges of these formulations are particularly common to currently marketed nasal salt products:

- **FEND1**: a 2×isotonic composition [0.12 M $CaCl_2$, 0.15 M NaCl] (1.29% CaCl2, 0.9% NaCl);

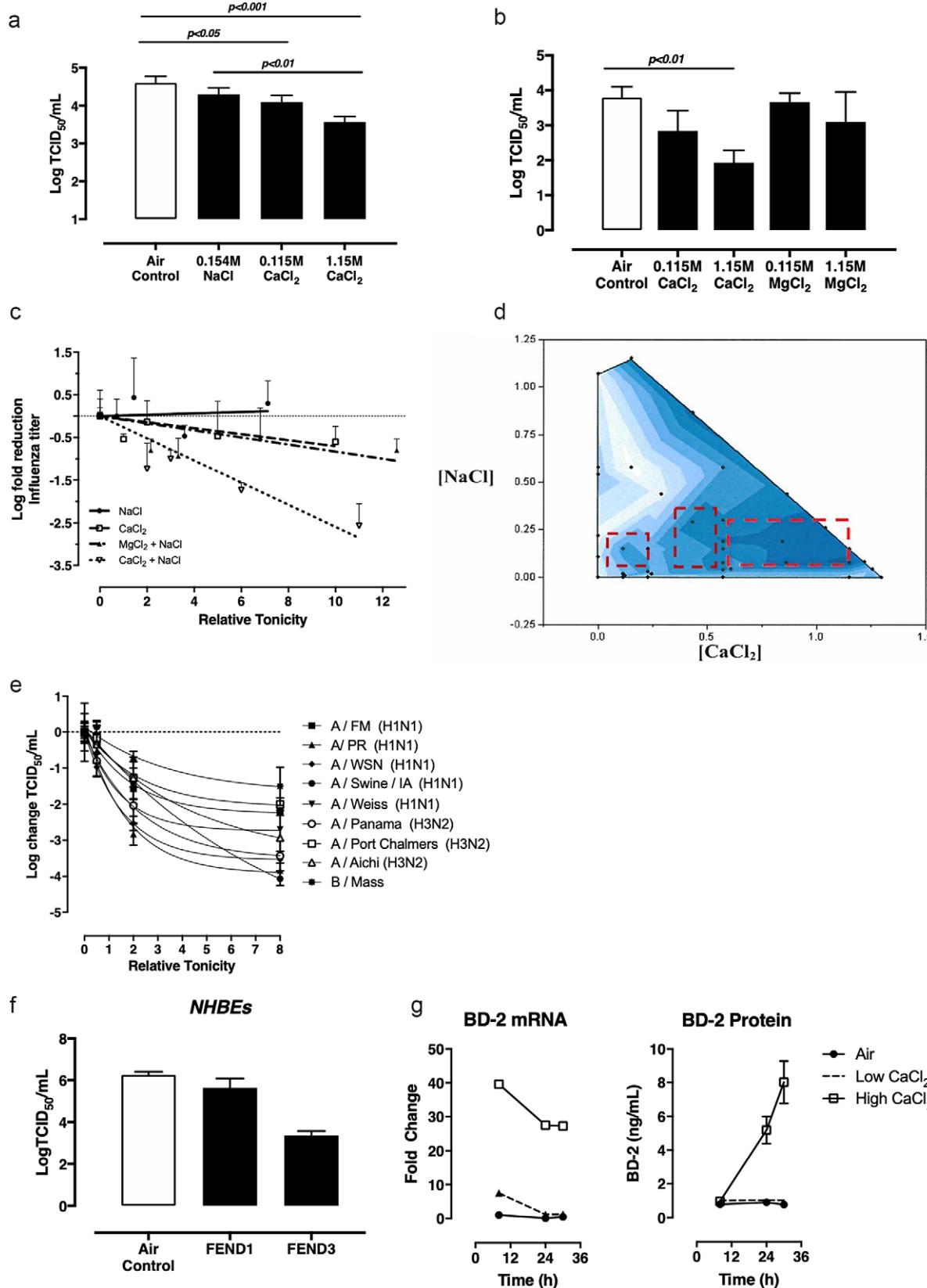

**Fig. 1.** The in vitro antiviral efficacy of calcium and sodium salt formulations. (*a*) Calcium chloride ($CaCl_2$) reduced influenza infectivity in a dose-dependent manner. Calu3 cells exposed to no formulation were used as a control and compared to Calu3 cells exposed to formulations of different concentrations of isotonic saline or $CaCl_2$ dissolved in isotonic saline. The of virus released by cells exposed to each aerosol formulation was quantified by 50% tissue culture infectious dose ($TCID_{50}$) assay. Bars represent the mean and standard deviation of triplicate wells for each condition. (*b*) Calu3 cells exposed to no formulation were used as a control and compared to Calu3 cells exposed to formulations of magnesium matched to that of calcium on a molar basis. The concentration of virus released by cells exposed to each aerosol formulation was quantified by $TCID_{50}$ assay. Bars represents the

- **FEND2**: a 4×isotonic composition [0.43 M CaCl$_2$, 0.05 M NaCl] (4.72% CaCl2 , 0.31% NaCl); and
- **FEND3**: an 8×isotonic composition [0.85 M CaCl$_2$, 0.11 M NaCl] (9.43% CaCl2 , 0.62% NaCl).

The first is a CaCl$_2$ augmentation of saline, the second near the salinity of sea water, and the third a composition with a salinity in the upper range of what has been observed as effective in previously published studies of antiviral nasal saline (Adam *et al.*, 1998; Dellanno *et al.*, 2009; Ramalingam *et al.*, 2019). These formulations were tested in Calu3 cells against a variety of Influenza A and Influenza B strains and shown to be effective in all cases with increased CaCl$_2$ concentration (Fig. 1*e*).

We sought to evaluate the calcium-mediated mechanism of action initially with normal human bronchial epithelial cells (NHBECs). We evaluated the effect of treatment on viral titer using the same ALI approach as described above for Calu3 cells. These data indicate that treatment of human bronchial epithelial cells with FEND formulations reduces viral titer (Fig. 1*f*). In addition, we evaluated mRNA expression of 26 genes following application of CaCl$_2$ formulations compositions (Supplementary Material). We found that 7 of the 26 genes consistently expressed three-fold or greater expression after treatment. These included: β-defensin 2 (BD2), interleukin 8 (IL-8), interleukin 6 (IL-6), 10-kDa interferon-γ–induced protein 10 (IP10), mucin 5AC, mucin 5B, and regulated on action, normal T expressed and secreted protein (RANTES). These results pointed to the ability of the calcium and sodium salt compositions to enhance innate immunity and notably expression of BD2 a powerful antimicrobial potent against viral pathogens including coronaviruses (Fig. 1g).

We performed a second set of experiments to assess calcium-induced retardation of microbial movement across simulated ALF mimetic systems as may relate to diminished infection rates in time-constrained viral infection systems such as the nose or the ALI systems in our studies. We first evaluated the speed of bacterial pathogen progress across the ALF mimetic. We topically applied by nebulizer aerosol saline or FEND1, then added the gram-positive and gram-negative pneumonia bacteria. The results are summarized in Fig. 2*a–e*. In all cases calcium interactions with the ALF mimetic increase the viscoelasticity of the ALF—as has been observed and explained elsewhere (Edwards *et al.*, 2004)—and lower rate of passage of the bacterial pathogens across ALF. This same effect was observed with influenza and rhinovirus as shown in Figs 2*f*,g, notably at concentrations of calcium in the range of the FEND2 and FEND3 compositions.

Optimal combinations of CaCl$_2$ and NaCl as typified by FEND1, FEND2, and FEND3 all show potential for antimicrobial activity by a combination of stimulating secretion of antimicrobial peptides and by slowing pathogen movement across ALF. We carried these compositions forward in animal and human studies.

## Small and large animal efficacy studies: pneumonia and influenza

To evaluate the therapeutic (antimicrobial) and hygienic (airway particle cleansing) properties of FEND in animals we examined mice and porcine models.

We first explored FEND antimicrobial efficacy in a mouse pneumonia model. Mice nasal and pulmonary exposure of an estimated 6.4 mg/kg was achieved by nebulizer either 2 h after (treatment) or 2 h before (prophylaxis) infection with *Streptococcus pneumoniae* (Serotype 3; ATCC 6303) with FEND1 or a saline control. As shown in Fig. 3*a* both in the case of treatment and prophylaxis bacterial burden is significantly lowered by the delivery of the calcium and sodium salts relative to the control. To verify that the efficacy of the FEND composition was driven by calcium per se and not by sodium, chloride, or the di-valency of calcium, we repeated the experiments with magnesium replacing CaCl$_2$ and with NaCl alone. Fig. 3*b* shows that no significant decline in bacterial burden follows nasal and pulmonary delivery of MgCl$_2$ relative to the untreated control, while aerosolization of CaCl$_2$ alone produces a significant decline in bacterial burden relative to NaCl alone.

We next studied the hygienic properties of FEND in a large animal (swine) model of influenza.

Piglets were randomly assigned to infected or naïve groups ($n = 8$ total; 4 per group) in each study and each group was housed in a separate clean room.

Following infection of untreated animals, naïve animals were exposed to the exhaled breath of the infected animals. Exposures were performed such that the infected animals were housed in the bottom of the exposure chamber and the naïve animals in the upper portion of the chamber to avoid contact transmission and allow only transmission via exhaled breath. Infected and naïve animals were each fitted with anesthesia masks and connected by >3″ of tubing. Once connected, naïve animals were exposed to the exhaled breath of the infected animals for 1 h/day at 48- and 72-h post infection.

In a separate group of piglets ($n = 8$ total; 4 per group), anesthetized animals were administered FEND1 aerosols at 48- and 72-h post infection to reduce airborne particle emission. In the first experiment, treatment largely eliminated infection as measured in terms of lung consolidation (Fig. 4*a*). In the second experiment, 10 min following treatment, naïve animals were exposed to the exhaled breath of the infected animals in the same manner as described above. Exhaled breath from untreated infected animals caused infection in naïve animals with a 75% transmission rate (Fig. 4*b*), suggesting transmission by bioaerosol.

Treatment by FEND1 aerosol (48 and 72 h post infection) with an estimated 0.5 mg/kg deposited dose reduced the infection rate in the exposed animals. FEND1 treatment of infected animals completely blocked transmission (0% transmission) to noninfected animals (Fig. 4*b*).

---

mean and standard deviation of triplicate wells for each condition. (*c*) The effect of multiple formulations of CaCl$_2$, magnesium chloride and NaCl on viral titers in influenza infected Calu3 cells, as quantified by TCID$_{50}$ assay. Bars represent standard deviation of triplicate wells for each condition. Data were analyzed statistically by one-way ANOVA and Turkey's multiple comparison post-test and indicated that CaCl$_2$ in a NaCl solution was the optimal combination of salts. (*d*) Data from multiple independent studies were pooled and the change in rate constant for each is plotted. The greatest reduction in rate constant was observed for calcium concentrations between 0.575 and 1.14 M and sodium concentration between 0.075 and 0.3 M. Three areas with darker the shades of blue are shown by the batched red boxes and indicate the greatest reduction in viral replication rate constant, optimizing the ratio of Ca$^+$ and Na$^+$ in the fast emergency nasal defense (FEND) formulations at an 8:1 ratio. (*e*) Anti-viral efficacy of the FEND formulation was evaluated across a range of influenza virus A and B strains. MOI 0.1 to 0.01. (*f*) Normal human bronchial epithelial cells (NHBECs) exposed to no formulation were used as a control and compared to cells exposed to FEND1 and FEND3. The concentration of virus (Influenza A/Panama/2007/99) released by cells exposed to each aerosol formulation was quantified by TCID$_{50}$ assay. Bars represent standard deviation of triplicate wells for each condition. Data were analyzed statistically by one-way ANOVA and Turkey's multiple comparison post-test. (*g*) NHBE donor cells were exposed to low and high CaCl$_2$ FEND formulations, then harvested 8, 24, and 30 h post treatment. BD-2 mRNA expression relative to the air (untreated) control was elevated at all time points after exposure to the high CaCl$_2$ formulation. In addition, there was a time-dependent increase in BD-2 protein in the apical surface wash between 4 and 30 hours after exposure in cells exposed to the high CaCl$_2$ FEND formulation relative to the air control.

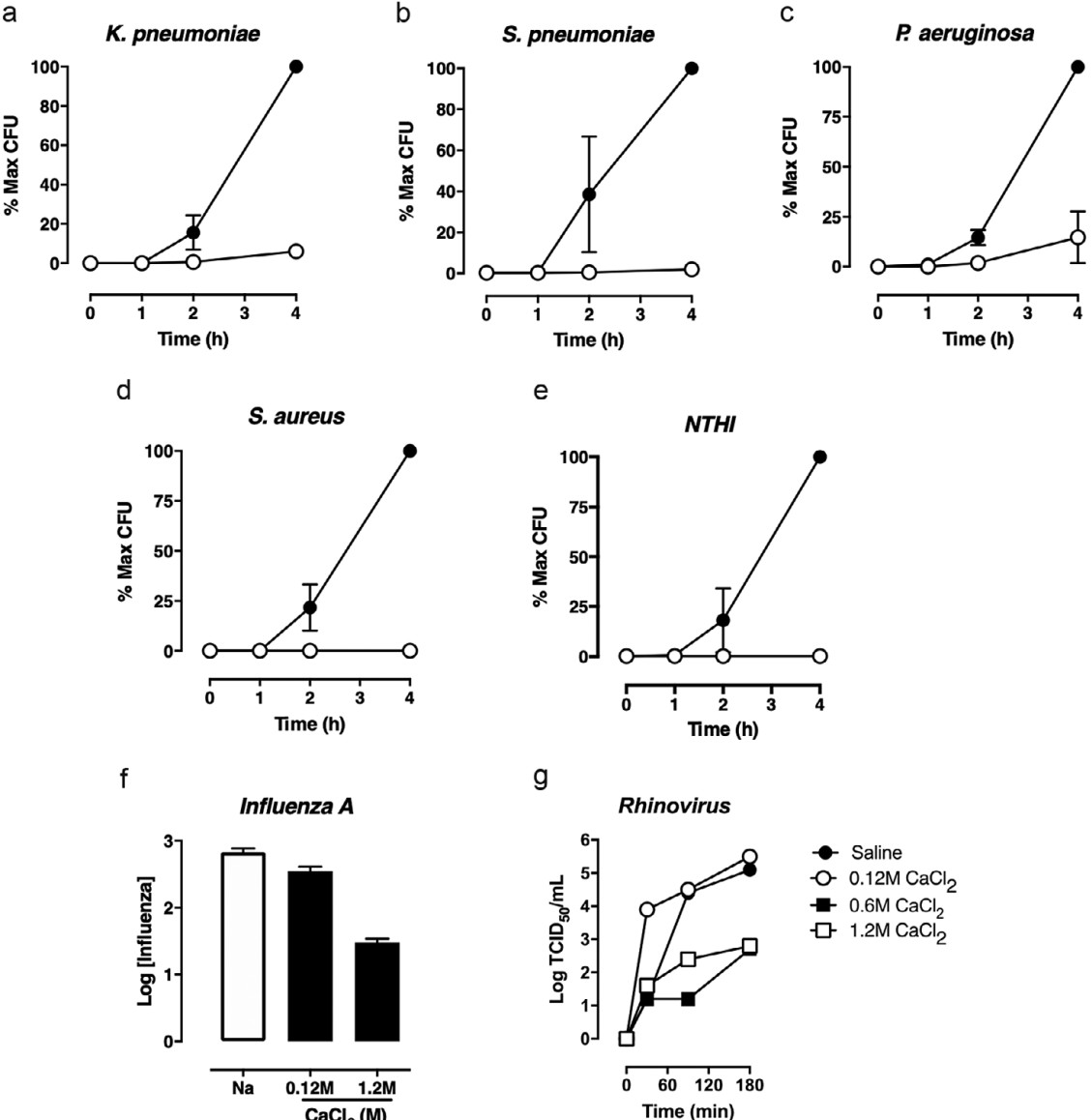

**Fig. 2.** Fast emergency nasal defense (FEND) formulations inhibit the movement of multiple bacterial and viral pathogens across mucus mimetic. (*a*–*e*) The number of bacteria and crossing 4% sodium alginate mucus mimetic over time (0–4 h) was measured. Data are expressed as the percent of the maximal titer for the saline control after 4 h. Mucus mimetic was treated topically with aerosol (saline [closed circles] or FEND1 [open circles]) and bacteria were added immediately post treatment (*n* = 3 independent experiments for *Klebsiella pneumoniae* and *Streptococcus pneumoniae*; n = 2 independent experiments for *Staphylococcus. aureus*, *Pseudomonas aeruginosa* and *Haemophilus influenzae*). Basolateral samples were serially diluted in saline and plated on blood or chocolate agar plates at each time point to quantify the number of bacteria. The percentage of the total number of bacteria passing through the mimetic in 4 h following saline exposure (maximum colony forming units [CFU] recovered from each assay) was calculated for each time point and the area under curve (AUC) of each curve was determined. In each case, the exposure of mimetic significantly reduced the number of bacteria crossing the mimetic compared to the control condition ($p < 0.05$ for *t* test of AUC between saline and FEND1 treatment group for each pathogen). (*f*–*g*). Sodium alginate mucus mimetic was treated with the indicated formulations and the movement of Influenza A/WSN/33/1 or Rhinovirus (Rv16) was assayed. Influenza was assayed from the basolateral buffer 4 h after treatment by quantitative polymerase chain reaction (qPCR) and Rhinovirus was assayed over time by 50% tissue culture infectious dose ($TCID_{50}$) assay. (*f*–*g*). Sodium alginate mucus mimetic was treated with the indicated formulations and the movement of Influenza A/WSN/33/1 or Rhinovirus (Rv16) was assayed. Influenza was assayed from the basolateral buffer 4 h after treatment by qPCR and Rhinovirus was assayed over time by $TCID_{50}$ assay. (*f*) Data depict the mean $\pm$ standard deviation of replicate runs of the qPCR reaction and are representative of three independent experiments. Data was analyzed by one-way ANOVA and Tukey's multiple comparison test (*$p < 0.05$ and **$p < 0.001$ compared to the saline control). (*g*) Data are representative of two independent experiments. The limit of detection of the $TCID_{50}$ assay was 1.2 $\log_{10}$ $TCID_{50}$/ml.

Given these results, we decided to pursue preclinical safety studies of both FEND1 and FEND2.

### Small and large animal toxicology studies

We performed 14-day studies of FEND1 and 28-day studies of FEND2 via daily nose only inhalation exposure in rats and daily nasal inhalation exposure in dogs. All aerosols had mass median aerodynamic diameters (MMAD) of <5 μm, to facilitate aerosol exposure to the entire respiratory tract, including nose and lungs.

In the 14-day repeat dose nonclinical safety studies, we exposed groups of animals to vehicle (saline) control and three doses of FEND1. For FEND1, no observed adverse effect level (NOAEL), expressed in terms of $CaCl_2$ dose, for *pulmonary* exposure in Sprague Dawley rats and in Beagle dogs, respectively were determined to be 13.5 and 8.77 mg/kg/day. While no adverse events were

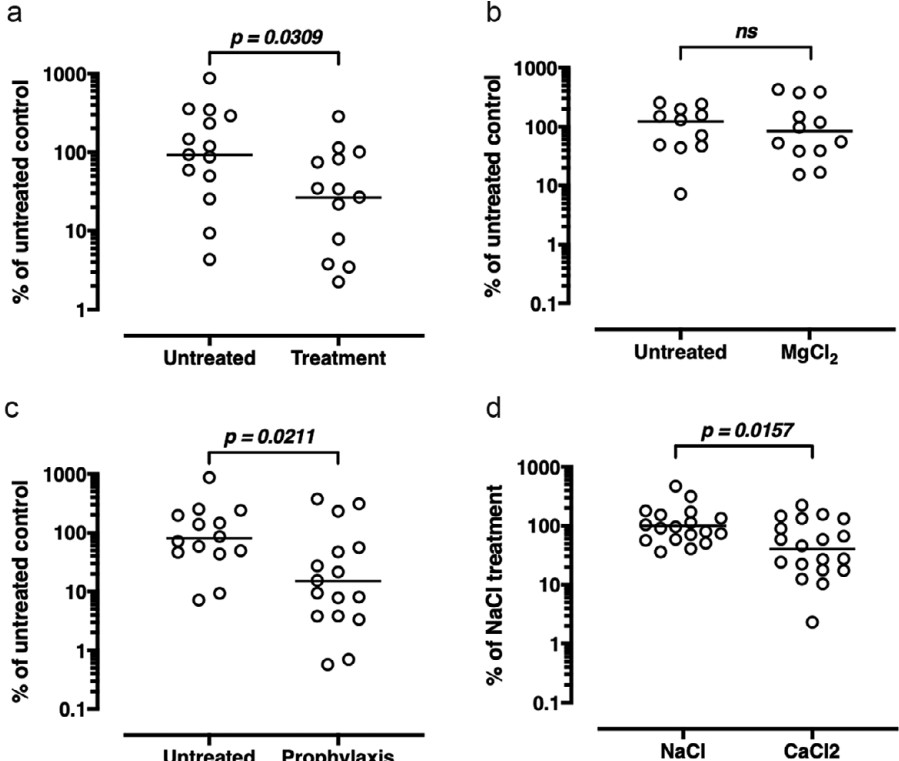

**Fig. 3.** Treatment and prophylaxis of pneumonia in mice by fast emergency nasal defense (FEND) aerosol. (*a*) Mice infected with *Streptococcus pneumoniae* and treated with CaCl$_2$-saline (FEND1) aerosol for 15 min 2 h after infection have less bacterial burden than untreated controls. (*b*) Mice infected with *S. pneumoniae* and pretreated with CaCl$_2$-saline (FEND1) aerosol for 15 min 2 h before infection have less bacterial burden than untreated controls. (*c*) Mice infected with *S. pneumoniae* and treated with MgCl$_2$ aerosol for 15 min 2 h before infection have a similar bacterial burden as untreated controls. (*d*) Mice infected with *S. pneumoniae* and pretreated with NaCl aerosol for 15 min 2 h before infection have a higher bacterial burden than animals pretreated with CaCl$_2$ aerosol. Pooled data from multiple experiments are shown. Each data point represents the data obtained from a single animal. The bar for each group represents the geometric mean of the group. The data were statistically analyzed using a Mann–Whitney *U* test (*ns* = not significant).

observed associated with nasal exposure, NOAELs for nasal exposure of FEND1 based on the pulmonary adverse events in Sprague Dawley rats and in Beagle dogs, respectively were determined to be 56.38 and 8.77 mg/kg/day.

In 28-day inhalation toxicology studies for FEND2, rats and dogs were administered vehicle (saline) control and three doses of FEND2. Again, no adverse events were observed for nasal exposure. NOAELs, expressed in terms of CaCl$_2$ dose of FEND2 and based on pulmonary adverse events, for both pulmonary and nasal exposure in Sprague Dawley rats and in Beagle dogs, respectively were determined to be approximately 10.2 and 18 mg/kg/day.

The results of our studies are summarized in Table 1 (see also Supplemental Materials).

Given these results, we decided to pursue human studies to evaluate human safety, and the potential of FEND to suppress exhaled bioaerosol when delivered to the entire respiratory system versus delivery preferentially to the nose.

### Human exhaled bioaerosol studies: pulmonary administration

A total of eight (*n* = 8) human subjects were administered ascending pulmonary doses of FEND1 (12.9, 38.7, and 77.4 mg CaCl$_2$), in the target efficacy range of 0.25–1.0 mg/CaCl$_2$/kg. Doses were delivered using a Pari LC hand-held nebulizer with mass median aerodynamic particle size of 3.1 μm suited to deposition through the respiratory tract and the deep lungs. All eight subjects received each dose and one dose of placebo (0.9% NaCl) on separate dosing days throughout the course of the trial. Safety assessments included

adverse event (AE) monitoring and concomitant medication as well as clinical laboratory evaluations, vital sign measurements, electrocardiogram (ECG), oxygen saturation, and pulmonary function test. All 8 subjects completed all 4 treatment days (1 placebo day and 3 days with escalating doses of the study drug).

There were no serious AEs (SAEs) reported in this study and no subjects were discontinued by the investigator due to an AE. A total of 17 treatment-emergent AEs (TEAEs) were reported by 7 (88%) of the 8 subjects dosed in this study, as summarized in the Supplemental Material. Of the AEs observed in pulmonary treated subjects, all were self-limited and resolved spontaneously without specific treatment.

We measured bioaerosol expiration (total particle counts per liter, and separately particles below and above 1 μm in size) in all of the subjects before and after treatment and compared with a placebo saline control.

Total exhaled particle counts per liter varied significantly between the individuals (Table 2). Some subjects exhaled many more than other subjects, consistent with previous observations of "super producer" subjects (Edwards *et al.*, 2004). Prior to placebo or treatment, five of the eight subjects exhaled on average $151 \pm 87$ particles per liter, while three of the individuals (subjects 2, 6, and 7) exhaled on average $1{,}201 \pm 1{,}715$ particles per liter. Suppression of bioaerosol by delivery of the FEND aerosol relative to placebo control occurred particularly at the high dose (see Supplemental Material for lower doses), as shown in Fig. 5*a*, with six of the eight subjects breathing out over the duration of 12 h post treatment, a significantly lower mean particle count per liter relative to placebo ($p < 0.05$). This included two of the three high-producer subjects

(6 and 7), who alone accounted for over 80% of the total exhaled bioaerosol particles of the group. Duration of effect was particularly strong between 2 and 6 h post treatment.

As further shown in Fig. 5b, suppression of bioaerosol expiration at the high dose of FEND1 is notably significant for bioaerosols in

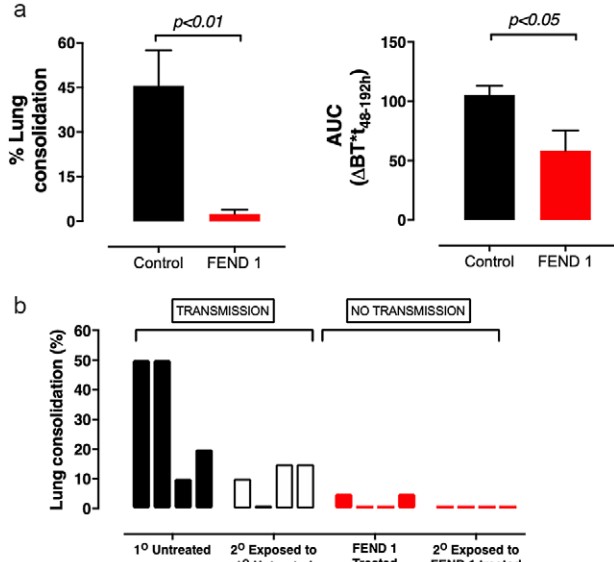

Fig. 4. Antiviral and anti-contagion efficacy of fast emergency nasal defense (FEND)1 against Influenza A infection in swine model. (a) Pooled data from replicate experiments (n = 8 control and n = 8 FEND1). Percent lung consolidation as a primary indicator of lung infection was significantly reduced (Unpaired t test p < 0.01) in animals treated with FEND1 relative to air treated controls. Similarly, reduced clinical illness, expressed as change in body temperature area under curve (AUC)$_{48-192h}$ (Unpaired t test p < 0.05). (b) Each bar represents the percent lung consolidation from the most affected lobe from a single animal. Naïve animals (n = 4 for each replicate) were secondarily exposed to the exhaled breath of primary infected untreated animals (black bars, n = 4) or primary infected treated animals (red bars, n = 4). Exhaled breath from the four infected swine was combined and delivered to individual naïve swine using a custom-designed exposure system designed to prevent direct contact transmission. Naïve swine were exposed to exhaled breath from infected animals for 1 h/day on day 2 and 3 post infection. Influenza transmission was observed between primary untreated animals and secondary animals (left black to gray bars). Primary infected treated animals (red bars) showed evidence of reduced infection rates and evidence of transmission was not observed (right red to light red bars).

the 300–1,000 nm range. These small particles represent the majority of bioaerosol particles (77%) measured in our study. Such submicron particles are not successfully captured by surgical masks (Leung et al., 2020), and are most prone to travel great distances, being too large to be particularly diffusive in the air and too small to settle by gravity.

The nose being a major compartment for viral infection, and with the high nasal prevalence of the human angiotensin-converting enzyme (ACE2) by which SARS-CoV-2 entry is mediated (Hou et al., 2020), we chose to develop a nasal delivery system to pursue a human nasal hygiene study and compare our pulmonary administration results with nasal delivery.

### FEND nasal delivery

We designed a hand-held nebulizer (Nimbus) for FEND capable of delivering nasal doses of around 1–2 mg CalCl$_2$ (see Supplementary Material). We integrated vibrating meshes (see Supplementary Material) with a 6 μm pore size to produce, on tipping of the device, an aerosol cloud with a particle size distribution optimal for delivery within the nose through natural nasal inspiration. The particle size distribution of the aerosol cloud reveals a median volume particle diameter of 9–10 μm (see Fig. 6a), an optimal size for nasal and upper airway deposition of aerosol following a natural tidal inspiration through the nose and with relatively uniform distribution of deposition from the anterior to the posterior of the nose (Calmet et al., 2019). The Nimbus particle size distribution is significantly smaller than that produced from a standard nasal pump spray (Fig. 6b).

On tipping (Fig. 6c), Nimbus produces $57 \pm 2$ mg within a 10 s actuation, after which power ceases until tipped back upright and again overturned. We designed the device to deliver a controlled dose of approximately 33 mg (i.e., 1.56 mg CalCl$_2$ for FEND2 or 0.43 mg CalCl$_2$ for FEND1) by filling an empty 6-oz glass with the cloud for the internally programmed 10-s actuation of the device and then inspiring the cloud directly from the glass into the nose (Fig. 6d). Uncontrolled dosing can also be achieved by creating the cloud before the nose and direct natural deep nasal inspiration (Fig. 6d).

We decided to pursue a human volunteer study with Nimbus to evaluate the effectiveness of FEND for suppressing exhaled aerosol particles following nasal administration in comparison to our observation of the effectiveness of FEND on pulmonary delivery.

**Table 1.** Results from 14-day rat and 14- or 28-day Beagle dog toxicology of FEND1 and FEND2 following aerosol exposure

| Formulation | FEND low strength | | FEND high strength | |
|---|---|---|---|---|
| Study | 14-day rodent | 14-day dog | 28-day rodent | 28-day dog |
| Study design | • 14-day with 14-day recovery | • 14-day with 14-day recovery | • 28-day with 14-day recovery | • 28-day with 14-day recovery |
| Achieved doses (Ca$^+$ mg/kg/day) | • 1.56, 4.42, and 18.4 | • 0.25, 0.69, and 1.9 | • 1.1, 3.7, and 11.8 | • 0.4, 1.5, and 6.5 |
| Outcomes | • No clinical observations or gross pathological changes<br>• No significant pathology in Low and Mid dose group:<br>• High dose group; Adverse findings in rat larynx, including inflammation, ulceration and calcification | • No clinical observations or gross pathological changes<br>• No ophthalmology or cardiovascular findings<br>• No significant pathology findings | • No clinical observations or gross pathological changes<br>• No significant pathology in Low and Mid dose group:<br>• High dose group: Microscopic findings in larynx, trachea and tracheal bifurcation related to irritation | • No clinical observations or gross pathological changes<br>• No ophthalmology or cardiovascular findings<br>• No significant pathology findings |

Rats and dogs tolerated daily inhalation administration at an estimated achieved dose up to several multiples of the efficacy dose as determined by the influenza ferret and swine studies. The clinical signs, body weights, food consumption, electrocardiograms, ophthalmology and clinical pathology parameters were unaffected by treatment. The studies were performed by ITR Laboratories in Canada.
Abbreviation: FEND, fast emergency nasal defense.

**Table 2.** Baseline exhaled particle counts from eight human subjects prior to placebo and three doses of FEND

| Subject | Placebo | Low | Medium | High |
|---|---|---|---|---|
| 1 | 193.24 | 185.19 | 256.96 | 355.23 |
| 2 | 166.22 | 3073.2 | 294.29 | 108.85 |
| 3 | 95.01 | 210.72 | 108.62 | 245.94 |
| 4 | 61.46 | 63.64 | 67.58 | 50.37 |
| 5 | 141.26 | 37.11 | 170 | 158.16 |
| 6 | 127.48 | 920.52 | 97.67 | 5941.78 |
| 7 | 1672.99 | 324.62 | 394.36 | 2101.53 |
| 8 | 63.28 | 206.56 | 281.29 | 73.87 |

Exhaled bioparticles per liter of the 8 human subjects prior to FEND or placebo treatment were counted by light scattering in four size bins, or categories (0.3–0.5, 0.5–1.0, 1.0–5.0, and >5.0 μm). Subjects were required to wear a nose clip and breathe into a mouthpiece for three deep breaths followed by approximately 3 min of targeted tidal breathing at each time point. Prior to each subject's measurement at each time point, the detection system was calibrated, providing background noise estimation for the filtration of the particles in the room environment.
Abbreviation: FEND, fast emergency nasal defense.

### *Human exhaled bioaerosol studies: nasal administration*

Ten healthy volunteers were recruited in St Augustine, Florida and Boston, MA. Each signed informed consent to participate in the several-hour nasal saline hygiene study. Five of these subjects were older than 65 years (70, 75, 82, 83, and 88) and five younger than 65 years (30, 40, 59, 60, and 63). Subjects with severe respiratory illnesses were excluded from the study, while two of the subjects (ages 30 and 63) were cigarette smokers.

All subjects began the study by breathing into an apparatus that measured expired aerosols. Following a baseline assessment of exhaled aerosol particle count subjects drew two deep nasal inspirations via Nimbus of FEND2. Subsequent to FEND administration, subjects breathed into the airborne particle detector at intervals for up to 6 h post dosing. Subjects also self administered a commercial (isotonic NaCl) simple saline cleansing spray (CVS Nasal Saline). Subsequent to nasal administration of the placebo control, subjects breathed into the airborne particle detector at intervals for up to 2 h.

Baseline exhaled particles per liter per subject age are shown in Fig. 7. Two of the ten subjects in the older age (>65) group exhaled very high numbers of particles per liter of air (24,088 ± 9,413 and 7,180 ± 1,250) (Fig. 7*a*) while the other eight (Fig. 7*b*) exhaled between approximately 10 and 1,200 particles per liter. In the latter group, two individuals (ages 30 and 63), were smokers. There is a strong correlation between high numbers of exhaled particles and age with the group older than 65 exhaling on average 6,641 particles per liter while the group younger than 65, on average, exhaling 440 particles per liter. In all subjects over 95% of the baseline exhaled particles were less than 1 μm in size with most smaller than 500 nm (Fig. 8*a*,*b*).

Following FEND administration, exhaled particle numbers diminished for up to several hours as shown in Fig. 8, presented on a log scale. This diminution relative to baseline is statistically significant ($p < 0.05$) for all 10 subjects. Duration of effect continued up to the last data point several (2–6) hours after administration for all of the subjects other than subjects B and E, each of whom were very small producers of particles. Administration of the simple saline control has a minor suppressive effect on exhaled particles for two of the subjects in the first hour following administration

while in the other subjects, we observed no suppressive effect (Fig. 9).

Using the lowest exhaled particle number following FEND administration as a measure of suppression effect, diminution of bioaerosol ranged from a low of 45% (subject age 75) to a high of 99% (subject ages 83 and 70), with overall suppression of aerosol for the group (99%) predominantly related to the dramatic effect of FEND on suppression among super producing individuals (subject ages 83 and 70).

### Discussion

Hypertonic $CaCl_2$ and NaCl solution delivered to the respiratory system appears to have potential as both hygienic and therapeutic biodefense against airborne pathogens. Hygienically, these physiological salts coat the surfaces of ALF to diminish breakup and clear away the submicron bioaerosol droplets (Fig. 5*c*) that are not effectively captured by masks (Fig. 4*b*). By potentially boosting natural immunological defenses—strengthening the barrier function of the ALF (Fig. 2) and promoting the secretion of β-defensin 2 from nasal and bronchial epithelial tissues (Fig. 1*g*)—these salts may also act therapeutically for antimicrobial prophylaxis or treatment.

While our results suggest that $CaCl_2$ and NaCl salt combinations may be therapeutically useful against bacterial and viral infections including influenza, rhinovirus, and pneumonia, our results point to an immediate hygienic value in the fight against any airborne infectious disease, including COVID-19, by cleaning the airways of the small airborne droplets that carry airborne infection.

Our finding that nasal inspiration of FEND in a group of 10 healthy human subjects reduces exhaled particles between 45 and 99% by way of an aerosol too large to penetrate the lower airways (Figs 8 and 9), suggests that the upper airways are a primary source of expired bioaerosol. The high velocity airstreams created during natural breathing (often reaching turbulent air flow conditions) in the trachea and main bronchi disturb the surfaces of ALF in the way of wind passing over the sea to generate sea mist (Wattanabe *et al.*, 2007). Such phenomena are highly sensitive to compositional variations in the underlying fluid, making exhaled bioaerosol a sensitive measure of ALF and introducing variability within and between subjects.

In our study, FEND substantially cleared away exhaled particles, most being less than 1 μm in size. That particles in the range of 300–500 nm were the most predominant observed in the exhaled breath of subjects can be explained by the fact that such particles are both too small to deposit in the lungs by gravity or inertia, once generated, and too large to be deposit by diffusion. These are the submicron particles most likely to remain suspended in the atmosphere essentially indefinitely. Possibly, most important in terms of their ability to transmit infection—and deposit on surfaces including airways of the infected or naive individuals—are those particles in the 500–1,000 nm range, a significant fraction of the exhaled particles of the super spreader individuals. These particles are also substantially eliminated by FEND treatment.

In our study, most of the airborne particles were exhaled from 2 of 10 "super producing" individuals. This super production of bioaerosol promotes the phenomenon of super spreaders (Stein, 2015). Super spreading events for COVID-19 have been reported in China (so-called patient 31), India (the Punjab outbreak), South Korea, and many other regions, and are suspected to be a primary mode of transmission of the disease (Kupferschmidt, 2020). Among key correlates of super spreading are suppressed immunity and infected lungs (Stein, 2015)—two particular vulnerabilities of the

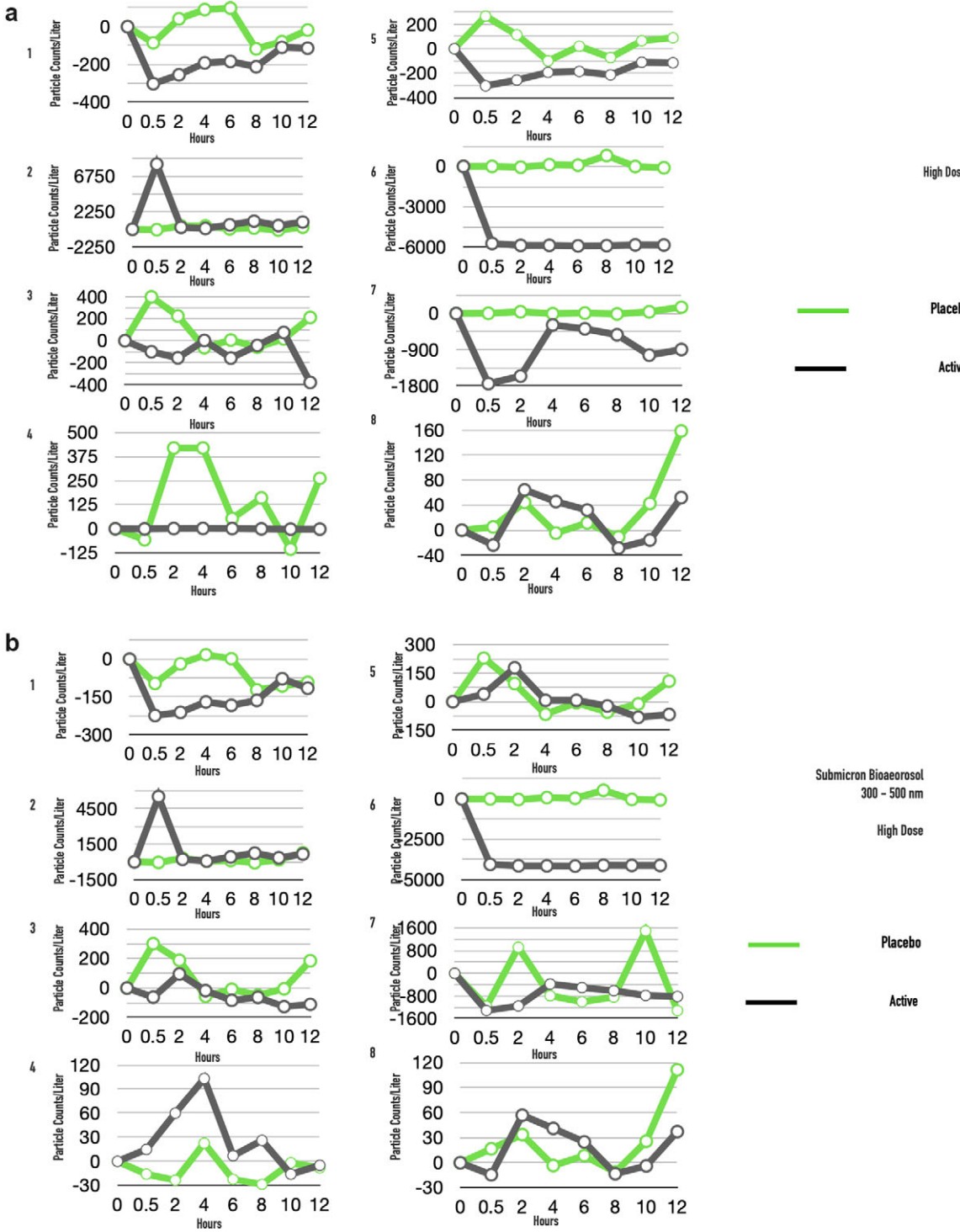

**Fig. 5.** Suppression of expired bioaerosol from human lungs following treatment by fast emergency nasal defense (FEND)1 at the high clinical trial dose. (*a*) Exhaled bioparticles per liter versus hours post dosing (relative to baseline) of the eight human subjects following FEND1 treatment or following saline placebo control. Bioparticle sampling from exhaled air was performed at the following time points: prior to dosing and at 0.5, 1, 2, 4, 6, 8, 10, and 12 h post dose for each period, and prior to release from the clinic on period 4, day 2. Each subject was treated with a placebo (saline) control and with each of the three doses of FEND1 at 24 h intervals. Data from each subject's high dose recording are compared with each subject's placebo control. (*b*) Expired bioparticles per liter (relative to baseline) in the size range of 300–500 nm from each of the human subjects are shown relative to placebo versus hour relative to dosing. Each subject was treated with a placebo (saline) control and with each of the three doses of FEND1 at 24-h intervals.

most aged. Indeed, older subjects in our study exhaled many more particles than younger subjects—suggesting the possibility that seniors, while among the most vulnerable to COVID-19 infection, may also be those most likely to spread the disease, and underscoring the extreme risk seniors face today in nursing homes.

As nasal hygiene, FEND is easy to administer (Fig. 6), rapid (one or two deep nasal inspirations) and lasts long (at least 6 h in those expiring the largest numbers of particles). It might be easily administered to individuals on entering environments, where they are likely to encounter others, including hospitals, nursing homes, prisons,

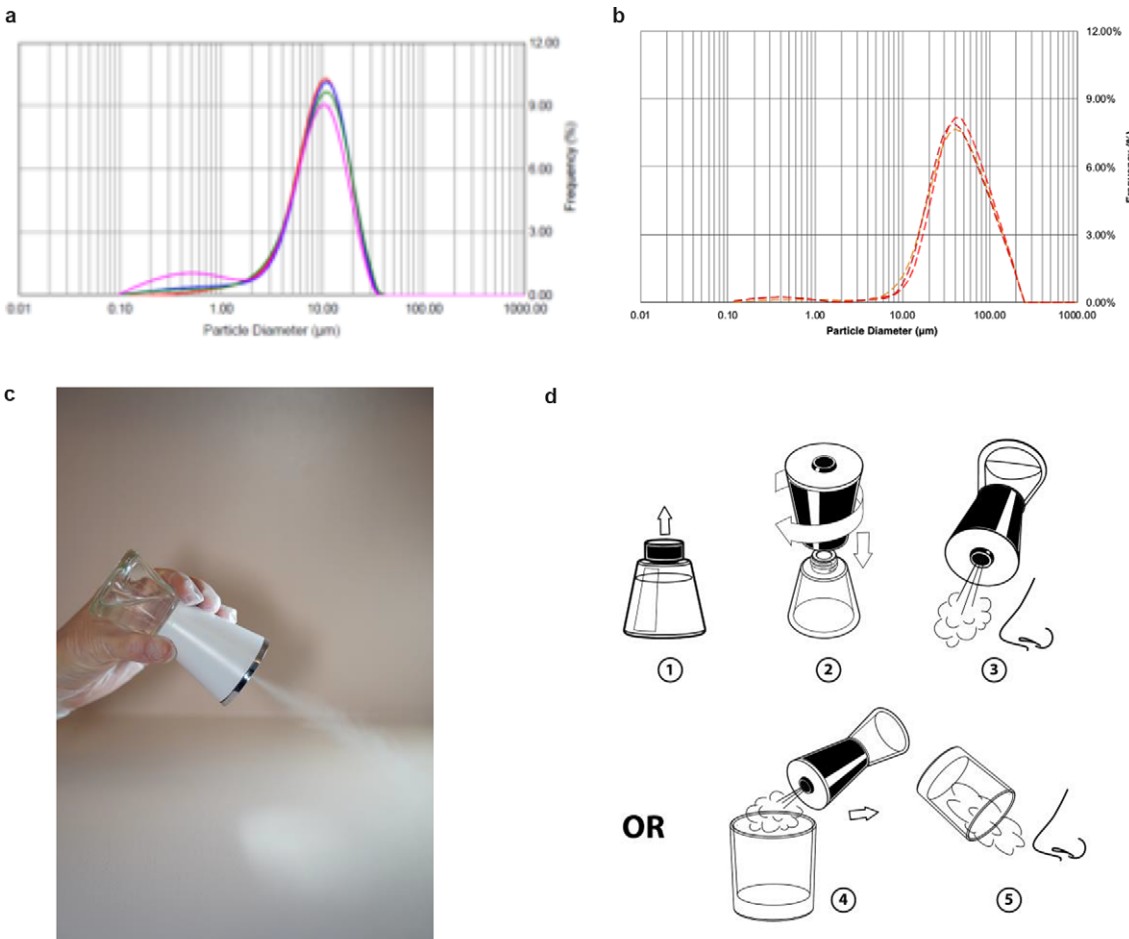

**Fig. 6.** Fast emergency nasal defense (FEND) delivery from nasal nebulizer (Nimbus). (*a*) The hand-held nebulizer discharged aerosol within an open-beam laser diffraction system (Malvern Spraytec) capable of measuring geometric size distributions of emitted droplets and particles. A fume extractor was used to draw the plume across the path of the laser. Triplicate measurements were performed twice on pure water yielding (9.5 μm average diameter) and on FEND2 composition yielding a mass mean aerodynamic diameter of 9.23 μm ± 0.60 and 8.84 μm ± 0.45. (*b*) Triplicate measurements were performed twice on a nasal pump spray triplicate measurements with FEND2 composition yielding a mass mean aerodynamic diameter of 107 μm. (*c*) Tipping of Nimbus actuated mesh vibration and generates an aerosol cloud for dosing. (*d*) FEND can be administered by Nimbus with a deep nasal inspiration either in an uncontrolled fashion before the nose or in a controlled fashion by containing the cloud in a glass.

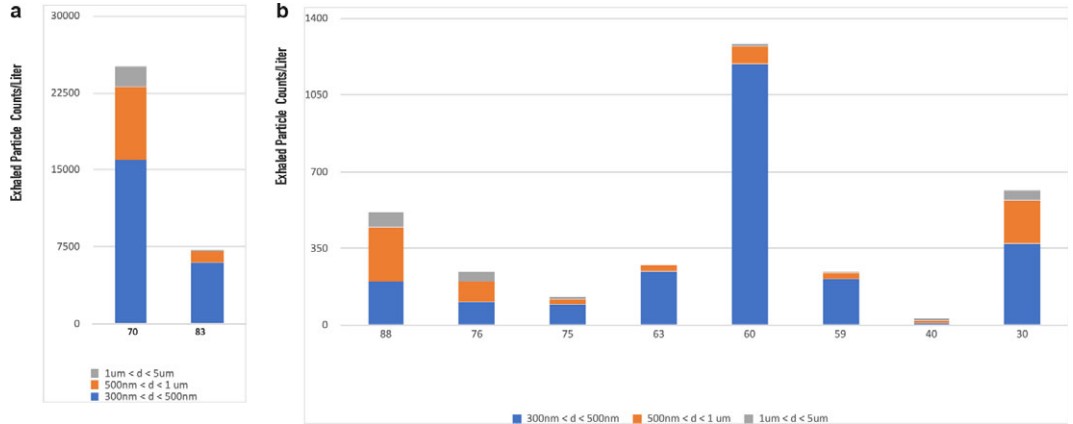

**Fig. 7.** Exhaled particles from the 10 human volunteers prior to fast emergency nasal defense (FEND) dosing. Exhaled particles per liter of air are shown within three size distributions —between 300 and 500 nm, 500 and 1,000 nm, and 1,000 and 5,000 nm. (*a*) Two of the human subjects (ages 63 and 70) exhaled greater than 25,000 and 7,000 particle per liter, respectively, the majority of these particles between 300 and 500 nm, and a large minority of the particles between 500 and 1,000 nm. (*b*) The other eight individuals breathed out on average several hundred particles per liter.

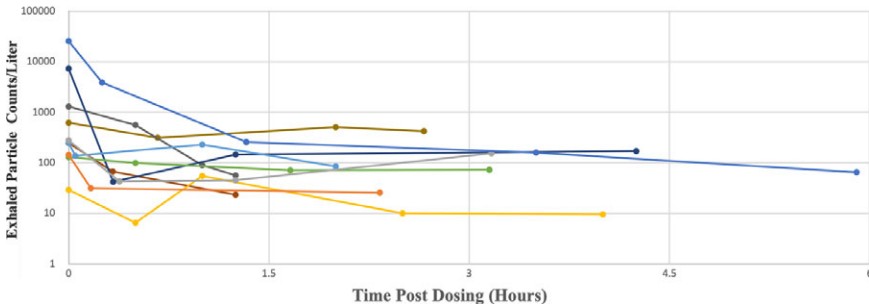

**Fig. 8.** Exhaled particles from the 10 human volunteers following fast emergency nasal defense (FEND) dosing. The exhaled particles per liter per human subject are shown on a log scale prior to dosing ($t = 0$) and at various times post dosing up to 6 h post dosing. In all cases statistically significant suppression of exhaled aerosol is observed while the effect is dramatically significant for the largest "super producing" subjects (ages 63 and 70), whose overall exhaled particle counts diminish more than 99% for 6 h following FEND nasal inspiration.

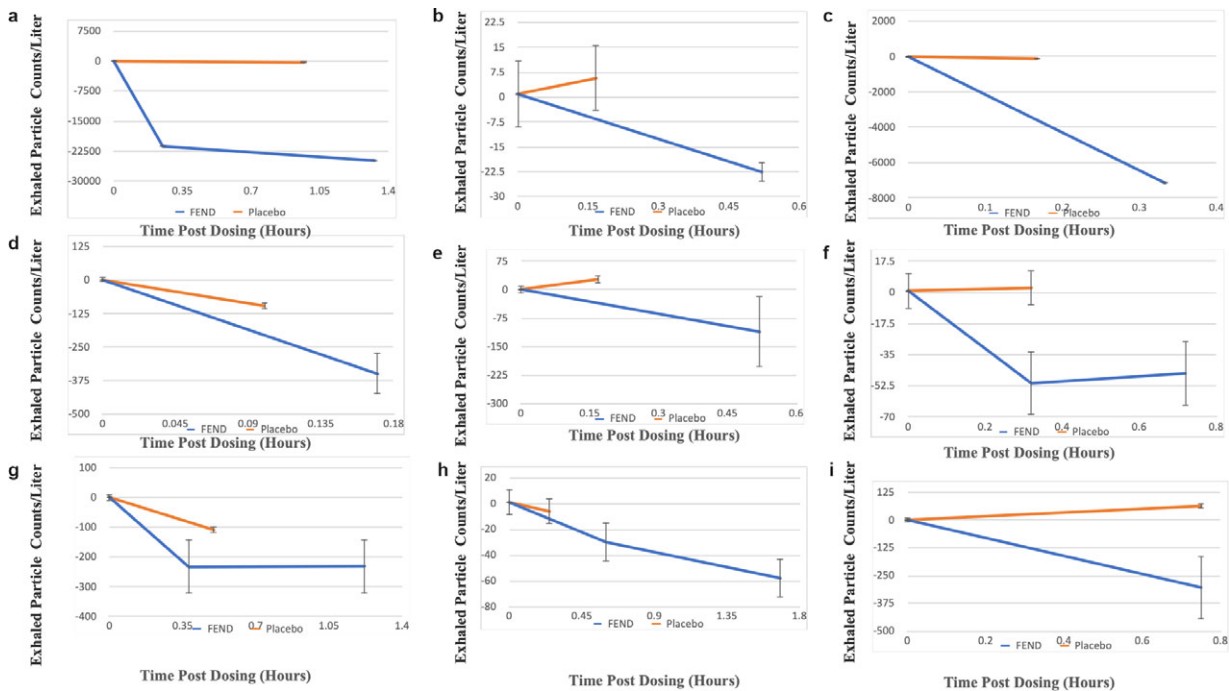

**Fig. 9.** Exhaled particles per subject following fast emergency nasal defense (FEND) dosing in comparison to the placebo control. All exhaled particles per liter (all sizes) are shown with standard error bars up to 1 h post dosing comparing the effects of FEND and isotonic saline (CVS Saline Spray) dosing on expired aerosol numbers For cases (*d*) and (*g*) the saline control shows significant suppression while for case (*f*) it show significant amplification. In all cases FEND suppresses exhaled aerosol counts relative to the control ($p < 0.05$) when comparisons are made between the closest time points of counts measured. The ages of the human subjects shown are: (*a*) 83, (*b*) 40, (*c*) 70, (*d*) 88, (*e*) 76, (*f*) 59, (*g*) 63, (*h*) 75, and (*i*) 30.

schools, offices, factories, stadia, restaurants, and museums. The use of FEND as an "invisible mask" supplement to traditional masks administered prior to close encounters with others in public and private spaces in order to clean the air of the small particles that masks do not block appears a prudent addition to current hygienic practices in the face of the COVID-19 pandemic.

More research is meanwhile needed to assess the consequences of calcium-enriched physiological salt nasal hygiene on airborne infection and transmission rates within environments at high-risk of COVID-19 and other airborne infection diseases. The therapeutic potential of these nasally and pulmonary delivery salts should also be explored beyond the scope of the in vitro and animal studies reported here.

## Materials and methods

### Study design

The goal of our studies was to explore the potential of a purely salt-based natural aerosol to promote innate immunity and dampen bioaerosol shedding with application to bacterial and viral airborne pathogens.

### In vitro model of influenza infection

Calu3 cells were cultured on permeable membranes (12 mm Transwells; 0.4 μm pore size, Corning Lowell, MA) until confluent. ALI

cultures were established by removing the apical media and culturing at 37°C/5% $CO_2$. Cells were cultured for >2 weeks at ALI before each experiment. Prior to each experiment the apical surface of each Transwell was washed 3× with 500 μl/well phosphate buffered saline (PBS) and the basolateral media (media on the bottom side of the Transwell) was replaced with 1.5 ml/well fresh media after aerosol exposure. NHBE cells were obtained from multiple donors from the University of North Carolina. Cells were cultured as describedin the Supplemental Materials. To establish ALI cultures, cells were cultured on permeable membranes similar to Calu3 cells. NHBE cells were cultured on Transwell permeable supports under ALI conditions for a minimum of 4 weeks before experiments.

Cells were exposed to nebulized aerosols using a customized sedimentation chamber that exposed triplicate wells of cells to the same aerosol. Aerosols were generated using Series 8900 nebulizers (Slater Labs). Following the delivery of formulations to cells, a second plate was exposed to the same formulations to quantify the delivery of total salt or calcium to cells. Following nebulization, each well of the second plate was washed with $dH_2O$ and salt concentration was assayed by osmometry.

Following treatment, cells were infected apically with 10 μl/well of Influenza A/WSN/33/1 at a multiplicity of infection of 0.1–0.01. Four hours after infection, the apical surfaces were washed with 500 μl/well PBS to remove excess formulation and unattached virus and cells were cultured for an additional 20 h at 37°C/5% $CO_2$. Twenty-four hours after infection, virus released onto the apical surface of infected cells was collected in 250 μl/well in plaque assay media (Ham's F12 plus supplements) and subsequently further diluted in 300 μl of plaque assay media before tittering. Assays with other influenza strains were conducted in the same manner. Influenza titers in recovered washes were determined by 50% tissue culture infectious dose ($TCID_{50}$) assay. Madin–Darby canine kidney cells (MDCK) cells were plated in 96-well flat bottom tissue culture plates (Costar) at 10,000 cells/well 24 h prior to use. Samples were serially diluted on MDCK cells in media containing TPCK-treated trypsin and cultured for 48–72 h at 37°C/5% $CO_2$. Infected wells were quantified by hemagglutinin assay using 0.5% chicken erythrocytes. Plates were incubated for 2 h at 4°C, positive wells were visually scored and $TCID_{50}$ was calculated according to the Spearman–Karber method. A rate constant was used as a means of normalizing data across studies such that the difference between placebo and active treated samples in a given study over the actual duration of bacterial of viral growth is calculated for each formulation in order to normalize across different studies.

### ALF mimetic

The apical surface of airway epithelium is lined by ALF that contains a mucus layer and a pericilliary layer. The ALF is rheologically complex and functions as a barrier for passage of pathogens and environmental particles protecting the epithelium. We developed a model system to mimic the mechanical (rheological) barrier of mucus following techniques previously described by Wattanabe *et al.* (2007). Our mucus mimetic was comprised of 4% alginate solution prepared from $2.0 \pm 0.0005$ g sodium alginate and 50 ml Milli-Q water (DI $H_2O$). The alginate samples were dissolved by adding a small amount to an aliquot of the water in a centrifuge tube, vortexing, and increasing step-wise the amounts of water and alginate. They were left to mix on the rotating wheel and stored at 4°C until homogenous. The alginate sample was used only when there was no visible bacterial growth. Since bacterial growth is known to lower the alginate's viscosity, a strain sweep was

also run at the beginning of each test. In all samples, $2.5 \leq \tan(\delta) \leq 3.0$ at 3 rad/s. The $\tan(\delta)$ variable is the ratio of the viscous to elastic modulus $G''/G'$, such that a decrease in $\tan(\delta)$ indicates a relative increase in elastic (vs. viscous) response. The increase in the viscous versus elastic response can be interpreted as an increase in the material's overall "stiffness" in oscillatory conditions. The $\tan(\delta)$ is measured using a rheometer with du Noüy ring to measure the tension between it and the surface of a sample using rotational motions with variable rate of oscillations and strain applied to a meniscus. The rate of oscillation is measured as rad/s over a range of 0.6283–15 rad/s and the strain is assessed over a range of 0.02–40%. The sodium alginate sample is poured into the sample dish and any large air bubbles drawn out by pipette. The du Noüy ring is lowered into the sample's bulk and raised again to a level slightly above the surface, so that a small, durable meniscus was visible. The strain sweep is run with a known oscillation rate to ensure that the viscosity is within the range of the normal mucus layer before treatment with the formulations and reassessment. See Supplemental Materials.

### BD-2 expression studies

For gene expression studies, total RNA was isolated from HBECs exposed to aerosolized formulations using the Qiagen RNeasy Plus Mini kit according to the manufacturer's Animal Cell Protocol. Expression levels of selected genes were quantified by quantitative polymerase chain reaction using the $\Delta\Delta Ct$ method and GAPDH as an internal reference for each sample (see Supplemental Materials). Secreted BD-2 protein levels were determined in apical washes of NHBE cells by ELISA (Peprotech Inc).

### Mice pneumonia studies

Specific pathogen-free female C57/BL6 mice (6–7 weeks, 16–22 g) were obtained from Charles River Laboratories and housed in the animal facility at Harvard School of Public Health according to their guidelines. Mice were given access to food and water *ad libitum.* For infections, *S. pneumoniae* (Serotype 3; ATCC 6303) were streaked onto blood agar plates and grown at 37°C plus 5% $CO_2$ overnight. Prior to infection, animals were anesthetized by intraperitoneal injection of a mixture of 50 mg/kg ketamine and 5 mg/kg xylazine. Single colonies of *S. pneumoniae* were resuspended in sterile saline to $OD_{600} = 0.3$ and subsequently diluted 1:4 in saline. Colloidal carbon was added to 1% and 50 μl of the resulting solution (~$1 \times 10^6$ colony forming units [CFU]) was instilled into the left lobe of anesthetized mice. Following infection, the bacterial titer of the inoculum was determined by serial dilution and plating on blood agar plates. After 24 h, mice were euthanized by exposure to isoflurane and the bacterial burden in lungs of infected animals was determined by plating serially diluted lung homogenate on blood agar plates.

A whole-body exposure system using a custom-designed high output nebulizer was utilized to deliver salt aerosols to a pie-chamber exposure system. Each pie chamber exposure chamber was modified such that a single tube delivered aerosol to a central manifold and ultimately to one of 11 mouse holding chambers via 4 inlet ports in each chamber. The total flow through the system was 11.7 L/min and animals were exposed to cationic aerosols for 15 min.

Mice were randomly assigned to different study groups on the day of the infections. Different aerosol exposure times relative to the time of infection were utilized to test the effect of aerosols in both prophylaxis and treatment regimens.

For each exposure, mice were loaded into a customized whole-body pie chamber system in which aerosols were delivered to a

central manifold and subsequently to each individual animal. Aerosol exposures consisted of 15 min exposures, which delivered an estimated dose of 6.4 mg/kg/day of $CaCl_2$. After 24 h of infection, animals were euthanized by isoflurane inhalation and the lungs were surgically removed and placed in sterile water. Lungs were homogenized using a glass mortar and pestle until no large tissue fragments were visible. CFU were enumerated by serially diluting lung homogenates in sterile water and plating on blood agar plates. Plates were incubated overnight at 37°C plus 5% $CO_2$ and CFU counted the following day.

### Swine influenza model

Normal, 4-week-old piglets were obtained from the Penn State Swine herd and allowed to acclimate for 1–2 weeks prior to study. Nasal swabs from piglets were screened for naturally occurring infections including PRRS (Porcine Reproductive Respiratory Syndrome), influenza, *Mycoplasma* and *Salmonella* and the piglets underwent a 1-week course of antibiotic therapy prior to influenza infection to limit confounding co-infection. Preinfection serology was performed to identify previous influenza exposure and only piglets with a hemagglutinin titer of <1:10 were used in studies. For influenza infections and aerosol treatments, animals were anesthetized with a mixture of tiletamine (Telazol, 2 mg/kg), zoleazepam (2 mg/kg), and xylazine (4 mg/kg). Animals were inoculated intranasally with swine influenza A (H1N1) virus grown for 48 h in 11-day-old embryonated eggs. The inoculum was prepared to deliver $10^7$ EID50 in 0.5 ml PBS administered intranasally. Animals were monitored twice daily for feed intake, rectal temperature, attitude, appetite, breathing rate, and evidence of dyspnea.

Anesthetized piglets were exposed to aerosols via a snout-only system via anesthesia masks in a custom-built holding chamber. Aerosol was generated from a custom-built high output ultrasonic nebulizer and delivered into a central manifold to which each individual animal was connected. Individual animals fitted with canine anesthesia masks were exposed to aerosol from the central manifold via one-way valves; animals similarly exhaled via one-way valves to maintain a dynamic flow. Animals were treated with FEND1 for 22 min on days 2 and 3 after infection. Body temperatures and clinical signs were recorded throughout the post-infection period. At day 9 post infection, animals were euthanized, lungs were removed and the percentage of the lung that showed consolidation as quantified.

For transmission studies, piglets were randomly assigned to infected or naive groups ($n = 8$ total; 4 per group) and each group was housed in a separate clean room. Following infection, anesthetized animals were treated with FEND1 on days 2 and 3 after infection as in treatment studies. Ten minutes following treatment, naïve animals were exposed to the exhaled breath of the infected animals. Infected and naïve animals were each fitted with anesthesia masks and connected by >3″ of tubing. Once connected, naïve animals were exposed to the exhaled breath of the infected animals for 1 h/day on days 2 and 3 after primary infection.

### Non-clinical safety studies

The study protocols were reviewed and assessed by the Animal Care Committee (ACC) of ITR, Montreal, QC, Canada. Animals were cared for in accordance with the principles outlined in the current "Guide to the Care and Use of Experimental Animals" as published by the Canadian Council on Animal Care and the "Guide for the Care and Use of Laboratory Animals", an NIH

publication. Studies were conducted in compliance with Good Laboratory Practice (GLP).

Sprague Dawley rats ($n = 10$/sex/group) were exposed to FEND1 or FEND2 by nose-only inhalation once daily for up to 120 min per exposure. Doses were achieved by exposing animals to different durations of each formulation. Control animals received isotonic saline for 120 min. Aerosol was produced using two Sidestream nebulizers in tandem. The airflow rate through the exposure system was monitored and recorded manually during the aerosol generation and was controlled using variable area flow meters. Determinations of aerosol concentration, particle size distribution, oxygen concentration, relative humidity, and temperature were performed on test and control atmosphere samples collected from a representative port of the exposure chamber. In the FEND1 study, rats were treated for 14 days at doses of 4.79, 13.5, and 56.4 $CaCl_2$/kg and in the FEND2 study, rats were treated for 28 days at doses of 3.05, 10.3, and 32.7 $CaCl_2$/kg. MMAD in the FEND1 study ranged from 1.3 to 2.0 μm with geometric standard deviations (GSD) of 1.63–2.03. In the FEND2 study, MMADs were 1.5–1.9 μm with GSD of 2.20–2.37.

In-life observations were made starting 1 week prior to dosing and continued through the dosing period. These included clinical observations, body weight, food consumption, ophthalmoscopy, clinical pathology, hematology, clinical chemistry, and urinalysis. At necropsy, gross pathology was conducted on internal exam and tissues were preserved in neutral buffered formalin for histopathological analysis and microscopic analysis of hematoxylin and eosin stained slides.

Beagle dogs ($n = 3$/sex/group) were exposed to FEND1 or FEND2 by orthonasal inhalation once daily for up to 120 min per exposure using a facemask exposure system. Aerosol generation and monitoring was conducted similar to the rat study, with adjustments made for the dosing system. In the FEND1 study, dogs were treated for 14 days at doses of 1.15, 3.24, and 8.77 mg $CaCl_2$/kg and in the FEND2 study, dogs were treated for 28 days at doses of 1.11, 4.16, and 18.0 $CaCl_2$/kg. In the FEND1 study, MMADs were between 0.9 and 1.2 and GSDs ranged from 1.67 to 1.97 across groups. In the FEND2 study, MMADs were 1.2–1.3 μm with GSD of 1.90–2.05. In-life observations with similar to the rat studies, with the additional inclusion of ECG on the first and last day of dosing.

### In vitro particle size determination

Emitted size distributions from the FEND Nasal Nebulizer and FEND Nasal Spray were determined via laser diffraction using the Spraytec spray analysis system (Malvern Panlytical Ltd, UK) in an open-bench configuration. The emitted size distributions of purified water and the FEND2 formulation were assessed from each delivery system. The delivery devices were affixed approximately 2″ from the measurement beam. Data collection occurred at a 1 kHz acquisition rate over the duration of the spray event, with reported results representing the time-averaged size distributions. All experimental conditions were assessed in triplicate.

### Human pulmonary administration

A Phase l, double-blind, placebo-controlled, randomized study was conducted to evaluate the safety and tolerability of single ascending doses of FEND1 and to assess the pattern of exhaled bioparticles following FEND1 inhalation. The trial enrolled eight healthy subjects who consented to receive a single dose of FEND1 or placebo (isotonic saline) once daily in four periods with four dosing sequence groups. Doses of 12.9, 38.7, or 77.4 mg $CaCl_2$ were administered in ascending

order with two subjects receiving placebo on each day of dosing. Dose escalation to the next dose level was only permitted if, in the opinion of the Principal Investigator, adequate safety and tolerability were demonstrated at the previous lower doses.

Subjects were confined to the clinic from the evening prior to the first dose until 24 h after the final dose in period 4. Subjects returned to the clinic approximately 7 days following the final dose, for follow-up safety assessments. Primary endpoints were clinical signs and symptoms from physical examination, AEs, laboratory safety (hematology, serum chemistry, and urinalysis), vital signs (blood pressure, heart rate, temperature, and respiratory rate), ECGs, oxygen saturation, and pulmonary function test. The secondary endpoint was exhaled bioparticle measurements.

Exhaled bioparticles were counted and measured for size distribution. Exhaled bioparticle samples were analyzed for number and size distribution using a proprietary bioaerosol expiration particle counter. Particles were counted and sized by light scattering into 4 size bins, or categories (0.3–0.5, 0.5–1.0, 1.0–5.0, and >5.0 μm). Exhaled breath profiles were collected and passed through a laser particle counter and a flow meter measured the full inhalation and exhalation airflow rate. Software combined these two measurements and calculated all relevant parameters. Primary measurements were total particle counts/liter, and the particle counts/liter that fell into the four size bins (bin cut points in μm): 0.3–0.5, 0.5–1.0, 1.0–5.0, and >5.0 counts/liter. Exhaled bioparticles were counted and measured for size distribution. Exhaled bioparticle samples were analyzed for number and size distribution using a proprietary bioaerosol expiration particle counter. Particles were counted and sized by light scattering into four size bins, or categories (0.3–0.5, 0.5–1.0, 1.0–5.0, and > 5.0 μm). Exhaled breath profiles were collected and passed through a laser particle counter and a flow meter measured the full inhalation and exhalation airflow rate. Software combined these two measurements and calculated all relevant parameters. Primary measurements were total particle counts/liter, and the particle counts/liter that fell into the four size bins (bin cut points in μm): 0.3–0.5, 0.5–1.0, 1.0–5.0, and >5.0 counts/liter.

### Human nasal administration

To establish the dose for our nasal administration human volunteer study, we determined a FEND efficacious dose level for bioaerosol mitigation per surface area of airway tissue of $0.00015 \, mg/cm^2$, based on a human lung surface area of $50 \, m^2$ based on the finding from our human pulmonary-administration study that inhalation of FEND1 produces significant suppression of expired bioaerosol at the highest total lung dose only. For nasal administration, this translates into a dose range from 25 μg $CalCl_2$ to 2 mg $CalCl_2$, with the lower end of the range based on our observed exhaled aerosol suppression and a standard nasal surface area of $181 \, cm^2$ and the upper end of the range based on our observed antimicrobial activity as observed in mice and pigs.

In our nasal administration human volunteer study exhaled particles were measured by a particle detector (Climet 450-t) designed to count airborne particles in the size range of 0.3–5 μm. The particle detector was connected to standard nebulizer tubing and mouthpiece that filters incoming air through a HEPA filter. Each standard nebulizer tubing and mouthpiece was removed from its sealed packaging before each subject prior to the subject's first exhaled particle detection. On subsequent counting maneuvers, the same mouthpiece and tubing was replaced into the particle counter system to insure oral hygiene. Subjects breathed at normal tidal breathing through a mouthpiece, while plugging their noses over at

least 2 min—beginning with one or two deep breaths. Over this time frame, particle counts per liter diminished from the ambient particle count to a lower number representing the particles emitted solely from the subject's airways. Once the lower plateau of particle counts was reached subjects continued to breathe normally. Three to six particle counts (average values assessed over 5 s) were then averaged to determine the mean exhaled particle count and standard error.

FEND was administered to each subject by tipping into an empty 6-oz glass. After 10 s actuation, Nimbus delivers approximately 33 mg (i.e., 1.56 mg $CalCl_2$) from the glass into the nose and upper airways. Dosing is accomplished by subjects immediately inspiring through the nose the cloud directly from the glass. Subsequent to FEND administration subjects breathed into the airborne particle detector at intervals from 10 min post dosing to 6 h post dosing.

**Open Peer Review.** To view the open peer review materials for this article, please visit http://doi.org/10.1017/qrd.2020.9.

**Supplementary Materials.** To view supplementary material for this article, please visit http://doi.org/10.1017/qrd.2020.9.

**Acknowledgments.** Our research was performed over a period of several years from our first experiments at Harvard, MIT and Penn State University—at Pulmatrix, and in a group of collaborating labs in the United States, Canada, England, and Northern Ireland. Many Pulmatrix scientists and technicians participated in the execution of these studies. Jérôme Edwards performed statistical analysis on the human bioaerosol data. Funding for all of the in vitro and in vivo pulmonary studies was provided by Pulmatrix, funding for all of the in vitro and in vivo nasal studies was provided by Sensory Cloud, and all FEND compositions have been licensed from Pulmatrix to Sensory Cloud.

**Conflict of interest.** DE is a shareholder and director of Sensory Cloud; AH is a shareholder and Scientific Advisory Board member of Sensory Cloud; RB is a shareholder and director of Pulmatrix; ML, WD, RC, DH, JP, BL and AC are shareholders of Pulmatrix. The authors declare no other conflict of interest.

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
