## [Reviewer Report]

*Comments to Author*: Edwards et al. reported a systematic, thorough and timely study of natural antimicrobial and anti-contagion against airborne pathogens using a simple and inexpensive device and a salt solution containing various concentrations of common salts NaCl and CaCl_2_. These salts are immediately available worldwide and affordable and the solution is easy to make. Such simple salt solution may save countless lives during the current COVID-19 pandemics. They cited 3 key previous studies as references for their current systematic studies, one reference is their own, Edwards et al 2004.

Edwards DA, et al (2004) Inhaling to mitigate exhaled bioaerosols. PNAS 101, 17383-17388.

Ramalingam S, et al (2018) Antiviral innate immune response in non-myeloid cells is augmented by chloride ions via an increase in intracellular hypochlorous acid levels. Scientific Reports 8(1).

Ramalingam S, et al (2019) A pilot, open labelled, randomized controlled trial of hypertonic saline nasal irrigation and gargling for the common cold. Scientific Reports 9(1).

These authors demonstrated that such common NaCl and CaCl_2_ salt solution is easy to make, to store and to make into nasal and pulmonary aerosols for immediate uses. Since the salt ingredients are easy to obtain and inexpensive, it can be quickly adopted by the world population including developing world population to combat the COVID-19 pandemics.

They carried out experiments, not only in tissue culture using Normal Human Bronchial Epithelial Cells (NHBE) to assay for its BD2 mRNA expression but also for BD2 protein production after exposure to CaCl_2_. They also used Calu3 cells against a variety of Influenza A and Influenza B strains for their NaCl and CaCl_2_ salt solution tests that are shown to be effective in all cases with increased CaCl_2_ concentration.

They systemically tested the simple salt aerosols in mice, rats, beagle dogs, pigs and 8 human voluntaries. They tested the salt aerosols on various pathogens including influenza A/WSN/33/1, rhinovirus (Rv16), and a range of gram-positive and gram-negative bacterial, Streptococcus pneumonia (Sterotype 4; TIGR4), Klebsiella pneumonia, Pseudomonas aeruginosa (PAO1), Staphylococcus aureus and non-typeable Haemophilus influenza (14P14H). Their results are sound and their claims are justified by their well-planned experiments and tests with very good controls.

Although they have not tested their salt solutions on current coronavirus Covid-19, it is presumed that such salt solution can be tried since it is harmless for human uses as they have already tested on human clinical trial voluntaries and analyzed and showed the data. The salt solution indeed has no harm to human during the small scale clinical trials.

Although they have not conducted large scale human clinical trial studies and there may be some unknown side effects that are only known in large scale studies, the benefit is significant outweigh the common salt side effect. Furthermore, there is no any effective vaccine available in near future, this reviewer agrees with their final statement “at a time when personal hygiene and antiseptic methods are at the forefront of daily living, it seems prudent to consider the immediate introduction of readily available nasal saline as part of that repertoire”.

There are some minor points:

1) Please move the explanation of FEND (fast emergency nasal defense) from late part of Results under FEND Nasal Delivery to first part of Introduction, so the readers can understand, from the beginning, what FEND means, rather read it later.

2) It is better to use the concentration of FEND in Results as millimolar, it is a more precise measurement than %. They can put % in ( ) after mM.

3) Please place A, B, C, D, E, F, G on each panel of Figure 1. In Figure 1G, please explain ~24 hours why the mRNA level goes down (left panel), but the protein level goes up (right panel). Is because the mRNA being degraded after 24 hours?

4) Figure 3, please label panels A, B, C, D.It is better to switch the locations of panels B and C, so it is consistent with other figures, namely, A and B are on the same horizonal level. There are no circles (number of mice) on MgCl_2_ on panel C. Please make it clear. If there are no differences between treated and nontreated mice, 3 circles representing 3 mice should be marked on MgCl_2_ part.

5) Figure 4, there are 3 panels, please use A, B, C to make them clear, since the Y axis for the left panel A is different from the right panel. They should be labelled as panel A and B.

6) Figure 5A, what are the thin black line and circle that reach to 6000? Is this some kind of control? The black line and circle are not represented in the colored side panel. What does it mean “expired bioaerosol”? Why “expired”?

7) Figure 5B, what is the Y-axis? Please label X-axis as hours. The numbers on Y-axis vary widely, patients 2 reaches ~6000, others only in 200-500, patient 4 reaches -6000, what do they mean? The figure legend is not clear.

8) Figure 5C, please label Y-axis. Please label X-axis as hours.Similar as Figure 5B, patients 2 reaches ~4500, others only in 200–500, patient 4 reaches -5000, what do they mean?

9) Figure 6, please use fine log scale for both panel A and B, rather only panel A, so the readers can immediately estimate the µm size. It is not easy to estimate µm size in panel B.

10) Supplementary Figure S1, please place a rule in centimeters, or a scale bar next to the device, so readers have a general idea how large the device is.

11) Supplementary Figure S2, it is better to add 2–3 steps (label steps A, B, C, D, etc) for the cartoon to demonstrate how this device is used. This reviewer did not understand it at first and asked someone else to explain how to use it.Ideally, no words are needed for the cartoon series, similar as he airplane emergence instructions, people from around the world can understand the cartoon immediately.

12) Supplementary Figure S3, please use the fine log scale for X-axis.

After they make these changes, this reviewer highly recommends expedite publication of this paper in QRB Discovery so the world population can benefit such simple, inexpensive and effective device to combat COVID-19 pandemics.

---

## [Reviewer Report]

*Comments to Author*: The manuscript by Edwards et al. studies the role of inhaled calcium and sodium salts to prevent disease transmission through exhaled aerosols. The manuscript includes a wide range of methods and pathogens, combining results from cell cultures and animal models, and a complete Phase I human study. The authors show that calcium chloride in saline (in three formulations FEND 1-3) reduces the influenza viral concentration in cell cultures, and slows down viruses and bacteria moving through a mucus mimetic. Furthermore, FEND has a prophylactic/treating effect on S. pneumoniae infection in mice, and also blocks aerosol spread of influenza between swine. The tests on human subjects show that the salts are safe and quite well tolerated, and for some of the subjects, the number of exhaled aerosol particles decrease.

The manuscript is of good quality and highly innovative. The cell/animal results show convincingly that inhaled salts have a significant effect slowing down or preventing viral/microbial infection. Thus, FEND has huge potential if developed into a simple and affordable first barrier of protection against airborne disease in humans. It is reasonable to continue human trials based on these results, I therefore recommend publication with very minor revisions.

Some points:

The authors should clarify the dosage (and method of delivery) of salts in the in vitro experiments.

“Rate constant” appears in the Figure 1 captions, but without sufficient explanation in the Methods section.

The sentence “In addition, mRNA expression of 26 genes following application of CaCl_2_ formulations compositions.” is hard to understand. If possible, data on these 26 genes should be listed, preferably in the SI.

“CANA” (appears once) should be clarified/changed into calcium-sodium.

“(X mg/kg)” appears once.

The colors and legend in Figure 5A do not match, and the -1800 particle change (Fig 5B, subject 7) does not match Fig 5A.

The sentence “Five of the eight subjects prior to placebo or treatment exhaled on average 151 +/− 87 particles per liter, while three of the individuals exhaled between 92 and 5942 particles, with a mean expiration of 1201 +/− 1715 particles per liter.” is somewhat difficult to understand, and Figure 5A is not very clear. If possible, Figure 5A should be replaced with a table. Also, the number of particles prior to placebo or treatment should be included in the table.

There is a huge variability in number of particles in Fig 5A, both between subjects, and also for the same subject on different days. The high dose data is presented in Fig 5B, but low and medium dose data should also be shown, preferably in SI, to give a better understanding of the natural variability. Also, most of the decrease in number of particles seems to come from subject 6. Maybe a stacked area chart summarizing all subjects could be used in the main text, while moving Fig 5 B-C to SI.

The use of the sign ≤ is mathematically incorrect in “In all samples, 2.5 ≤ tan (δ) ≥ 3.0 at 3 rad/s.” Also, the strain sweep (rheometer?) could be explaned in the methods section.

In the section “Non-clinical Safety Studies”, should “8.77 CaCl_2_/kg” be “8.77 mg CaCl_2_/kg”?

---

## [Reviewer Report]

*Comments to Author*: Reviewer #1: Edwards et al. reported a systematic, thorough and timely study of natural antimicrobial and anti-contagion against airborne pathogens using a simple and inexpensive device and a salt solution containing various concentrations of common salts NaCl and CaCl_2_. These salts are immediately available worldwide and affordable and the solution is easy to make. Such simple salt solution may save countless lives during the current COVID-19 pandemics. They cited 3 key previous studies as references for their current systematic studies, one reference is their own, Edwards et al 2004.

Edwards DA, et al (2004) Inhaling to mitigate exhaled bioaerosols. PNAS 101, 17383-17388.

Ramalingam S, et al (2018) Antiviral innate immune response in non-myeloid cells is augmented by chloride ions via an increase in intracellular hypochlorous acid levels. Scientific Reports 8(1).

Ramalingam S, et al (2019) A pilot, open labelled, randomized controlled trial of hypertonic saline nasal irrigation and gargling for the common cold. Scientific Reports 9(1).

These authors demonstrated that such common NaCl and CaCl_2_ salt solution is easy to make, to store and to make into nasal and pulmonary aerosols for immediate uses.Since the salt ingredients are easy to obtain and inexpensive, it can be quickly adopted by the world population including developing world population to combat the COVID-19 pandemics.

They carried out experiments, not only in tissue culture using Normal Human Bronchial Epithelial Cells (NHBE) to assay for its BD2 mRNA expression but also for BD2 protein production after exposure to CaCl_2_. They also used Calu3 cells against a variety of Influenza A and Influenza B strains for their NaCl and CaCl_2_ salt solution tests that are shown to be effective in all cases with increased CaCl_2_ concentration.

They systemically tested the simple salt aerosols in mice, rats, beagle dogs, pigs and 8 human voluntaries. They tested the salt aerosols on various pathogens including influenza A/WSN/33/1, rhinovirus (Rv16), and a range of gram-positive and gram-negative bacterial, Streptococcus pneumonia (Sterotype 4; TIGR4), Klebsiella pneumonia, Pseudomonas aeruginosa (PAO1), Staphylococcus aureus and non-typeable Haemophilus influenza (14P14H). Their results are sound and their claims are justified by their well-planned experiments and tests with very good controls.

Although they have not tested their salt solutions on current coronavirus Covid-19, it is presumed that such salt solution can be tried since it is harmless for human uses as they have already tested on human clinical trial voluntaries and analyzed and showed the data. The salt solution indeed has no harm to human during the small scale clinical trials.

Although they have not conducted large scale human clinical trial studies and there may be some unknown side effects that are only known in large scale studies, the benefit is significant outweigh the common salt side effect. Furthermore, there is no any effective vaccine available in near future, this reviewer agrees with their final statement “at a time when personal hygiene and antiseptic methods are at the forefront of daily living, it seems prudent to consider the immediate introduction of readily available nasal saline as part of that repertoire”.

There are some minor points:

1) Please move the explanation of FEND (fast emergency nasal defense) from late part of Results under FEND Nasal Delivery to first part of Introduction, so the readers can understand, from the beginning, what FEND means, rather read it later.

2) It is better to use the concentration of FEND in Results as millimolar, it is a more precise measurement than %. They can put % in ( ) after mM.

3) Please place A, B, C, D, E, F, G on each panel of Figure 1. In Figure 1G, please explain ~24 hours why the mRNA level goes down (left panel), but the protein level goes up (right panel). Is because the mRNA being degraded after 24 hours?

4) Figure 3, please label panels A, B, C, D.It is better to switch the locations of panels B and C, so it is consistent with other figures, namely, A and B are on the same horizonal level. There are no circles (number of mice) on MgCl_2_ on panel C. Please make it clear. If there are no differences between treated and nontreated mice, 3 circles representing 3 mice should be marked on MgCl_2_ part.

5) Figure 4, there are 3 panels, please use A, B, C to make them clear, since the Y axis for the left panel A is different from the right panel. They should be labelled as panel A and B.

6) Figure 5A, what are the thin black line and circle that reach to 6000? Is this some kind of control? The black line and circle are not represented in the colored side panel. What does it mean “expired bioaerosol”? Why “expired”?

7) Figure 5B, what is the Y-axis? Please label X-axis as hours. The numbers on Y-axis vary widely, patients 2 reaches ~6000, others only in 200-500, patient 4 reaches -6000, what do they mean? The figure legend is not clear.

8) Figure 5C, please label Y-axis. Please label X-axis as hours.Similar as Figure 5B, patients 2 reaches ~4500, others only in 200-500, patient 4 reaches -5000, what do they mean?

9) Figure 6, please use fine log scale for both panel A and B, rather only panel A, so the readers can immediately estimate the µm size. It is not easy to estimate µm size in panel B.

10) Supplementary Figure S1, please place a rule in centimeters, or a scale bar next to the device, so readers have a general idea how large the device is.

11) Supplementary Figure S2, it is better to add 2–3 steps (label steps A, B, C, D, etc) for the cartoon to demonstrate how this device is used. This reviewer did not understand it at first and asked someone else to explain how to use it.Ideally, no words are needed for the cartoon series, similar as he airplane emergence instructions, people from around the world can understand the cartoon immediately.

12) Supplementary Figure S3, please use the fine log scale for X-axis.

After they make these changes, this reviewer highly recommends expedite publication of this paper in QRB Discovery so the world population can benefit such simple, inexpensive and effective device to combat COVID-19 pandemics.

Reviewer #2: The manuscript by Edwards et al. studies the role of inhaled calcium and sodium salts to prevent disease transmission through exhaled aerosols. The manuscript includes a wide range of methods and pathogens, combining results from cell cultures and animal models, and a complete Phase I human study. The authors show that calcium chloride in saline (in three formulations FEND 1-3) reduces the influenza viral concentration in cell cultures, and slows down viruses and bacteria moving through a mucus mimetic. Furthermore, FEND has a prophylactic/treating effect on S. pneumoniae infection in mice, and also blocks aerosol spread of influenza between swine. The tests on human subjects show that the salts are safe and quite well tolerated, and for some of the subjects, the number of exhaled aerosol particles decrease.

The manuscript is of good quality and highly innovative. The cell/animal results show convincingly that inhaled salts have a significant effect slowing down or preventing viral/microbial infection. Thus, FEND has huge potential if developed into a simple and affordable first barrier of protection against airborne disease in humans. It is reasonable to continue human trials based on these results, I therefore recommend publication with very minor revisions.

Some points:

The authors should clarify the dosage (and method of delivery) of salts in the in vitro experiments.

“Rate constant” appears in the Figure 1 captions, but without sufficient explanation in the Methods section.

The sentence “In addition, mRNA expression of 26 genes following application of CaCl_2_ formulations compositions.” is hard to understand. If possible, data on these 26 genes should be listed, preferably in the SI.

“CANA” (appears once) should be clarified/changed into calcium-sodium.

“(X mg/kg)” appears once.

The colors and legend in Figure 5A do not match, and the -1800 particle change (Fig 5B, subject 7) does not match Fig 5A.

The sentence “Five of the eight subjects prior to placebo or treatment exhaled on average 151 +/− 87 particles per liter, while three of the individuals exhaled between 92 and 5942 particles, with a mean expiration of 1201 +/− 1715 particles per liter.” is somewhat difficult to understand, and Figure 5A is not very clear. If possible, Figure 5A should be replaced with a table. Also, the number of particles prior to placebo or treatment should be included in the table.

There is a huge variability in number of particles in Fig 5A, both between subjects, and also for the same subject on different days. The high dose data is presented in Fig 5B, but low and medium dose data should also be shown, preferably in SI, to give a better understanding of the natural variability. Also, most of the decrease in number of particles seems to come from subject 6. Maybe a stacked area chart summarizing all subjects could be used in the main text, while moving Fig 5 B-C to SI.

The use of the sign ≤ is mathematically incorrect in “In all samples, 2.5 ≤ tan (δ) ≥ 3.0 at 3 rad/s.” Also, the strain sweep (rheometer?) could be explaned in the methods section.

In the section “Non-clinical Safety Studies”, should “8.77 CaCl_2_/kg” be “8.77 mg CaCl_2_/kg”?

---

## [Reviewer Report]

*Comments to Author*: Referee report

Edwards et al. A New Natural Defense Against Airborne Pathogens

This is a revised manuscript that combined previously 2 manuscripts: 1) A Natural Antimicrobial and Anti-Contagion Against Airborne Pathogens, and 2) A natural hygiene for cleansing the air of the exhaled particles face masks do not stop.

The first manuscript is a very comprehensive study combining biochemistry, biophysics, molecular biology and animal studies of FEND, a simple but effective solution containing CaCl_2_/NaCl, and the second manuscript reports its clinical applications to human subjects.FEND is shown to be effective in human tests against spreading infections.

Their systematic studies spanning a wide range of areas including human that should have great benefit to prevent pandemic of COVID-19 around the world, especially in the developing countries since May 2020. The coronavirus infection rate has lately increased around the world alarmingly, particularly in the more vulnerable populations in the world.

The authors have already made corrections and addressed the concerns raised in both manuscripts by the reviewers. This combined manuscript is now in good standing to be published without further delay. This reviewer highly recommends expedite its publication. The sooner it is published, the sooner FEND solution can be used worldwide, the sooner it can prevent further spread coronavirus infections, and the more lives will be saved. The time is essence!

The Economist warned on July 3, 2020 the worst is yet to come for COVID-19 around the world. “The worst is to come. Based on research in 84 countries, a team at the Massachusetts Institute of Technology reckons that, for each recorded case, 12 go unrecorded and that for every two covid-19 deaths counted, a third is misattributed to other causes. Without a medical breakthrough, it says, the total number of cases will climb to 200m-600m by spring 2021. At that point, between 1.4m and 3.7m people will have died. Even then, well over 90% of the world’s population will still be vulnerable to infection—more if immunity turns out to be transient.”

---

## [Reviewer Report]

*Comments to Author*: After combining two manuscripts, “A New Natural Defense Against Airborne Pathogens” by Edwards et al. is of good quality and high potential scientific impact. The concept of reducing exhaled bioaerosol levels by inhaling simple salts could, in the near future, be developed into effectively blocking disease transmission when masks are unavailable or impractical. The single most interesting result is actually that calcium chloride apparently prevented lung consolidation in influenza infected piglets. Whatever the cause for the above, accounting for the effect of aerosols on the spread of Covid-19, I highly recommend publishing.

---

## [Reviewer Report]

*Comments to Author*: Reviewer #1: Referee report

Edwards et al. A New Natural Defense Against Airborne Pathogens

This is a revised manuscript that combined previously 2 manuscripts: 1) A Natural Antimicrobial and Anti-Contagion Against Airborne Pathogens, and 2) A natural hygiene for cleansing the air of the exhaled particles face masks do not stop.

The first manuscript is a very comprehensive study combining biochemistry, biophysics, molecular biology and animal studies of FEND, a simple but effective solution containing CaCl_2_/NaCl, and the second manuscript reports its clinical applications to human subjects. FEND is shown to be effective in human tests against spreading infections.

Their systematic studies spanning a wide range of areas including human that should have great benefit to prevent pandemic of COVID-19 around the world, especially in the developing countries since May 2020. The coronavirus infection rate has lately increased around the world alarmingly, particularly in the more vulnerable populations in the world.

The authors have already made corrections and addressed the concerns raised in both manuscripts by the reviewers. This combined manuscript is now in good standing to be published without further delay. This reviewer highly recommends expedite its publication. The sooner it is published, the sooner FEND solution can be used worldwide, the sooner it can prevent further spread coronavirus infections, and the more lives will be saved. The time is essence!

The Economist warned on July 3, 2020 the worst is yet to come for COVID-19 around the world. “The worst is to come. Based on research in 84 countries, a team at the Massachusetts Institute of Technology reckons that, for each recorded case, 12 go unrecorded and that for every two covid-19 deaths counted, a third is misattributed to other causes. Without a medical breakthrough, it says, the total number of cases will climb to 200m–600m by spring 2021. At that point, between 1.4m and 3.7m people will have died. Even then, well over 90% of the world’s population will still be vulnerable to infection—more if immunity turns out to be transient.”

Reviewer #2: After combining two manuscripts, “A New Natural Defense Against Airborne Pathogens” by Edwards et al. is of good quality and high potential scientific impact. The concept of reducing exhaled bioaerosol levels by inhaling simple salts could, in the near future, be developed into effectively blocking disease transmission when masks are unavailable or impractical. The single most interesting result is actually that calcium chloride apparently prevented lung consolidation in influenza infected piglets. Whatever the cause for the above, accounting for the effect of aerosols on the spread of Covid-19, I highly recommend publishing.